# Zero-Shot Reinforcement Learning from Low Quality Data

**Scott Jeen**
University of Cambridge
srj38@cam.ac.uk

**Tom Bewley**
University of Bristol
tomdbewley@gmail.com

**Jonathan M. Cullen**
University of Cambridge
jmc99@cam.ac.uk

## Abstract

Zero-shot reinforcement learning (RL) promises to provide agents that can perform *any* task in an environment after an offline, reward-free pre-training phase. Methods leveraging successor measures and successor features have shown strong performance in this setting, but require access to large heterogenous datasets for pre-training which cannot be expected for most real problems. Here, we explore how the performance of zero-shot RL methods degrades when trained on small homogeneous datasets, and propose fixes inspired by *conservatism*, a well-established feature of performant single-task offline RL algorithms. We evaluate our proposals across various datasets, domains and tasks, and show that conservative zero-shot RL algorithms outperform their non-conservative counterparts on low quality datasets, and perform no worse on high quality datasets. Somewhat surprisingly, our proposals also outperform baselines that get to see the task during training. Our code is available via the project page https://enjeeneer.io/projects/zero-shot-rl/.

## 1 Introduction

Today's large pre-trained models generalise impressively to unseen vision [70] and language [9] tasks, but not to sequential decision-making problems. Zero-shot reinforcement learning (RL) attempts to correct this, asking, informally: can we pre-train an agent on a dataset of reward-free transitions such that it can perform *any* downstream task in an environment? Recently, methods leveraging successor features [5, 7] and successor measures [6, 82] have emerged as viable zero-shot RL candidates, returning near-optimal policies for many unseen tasks [83].

These works have assumed access to a large heterogeneous dataset of transitions for pre-training. In theory, such datasets could be curated by highly-exploratory agents during an upfront data collection phase [32, 10, 18, 62, 63, 35, 48]. However, in practice, deploying such agents in real systems can be time-consuming, costly or dangerous. To avoid these downsides, it would be convenient to skip the data collection phase and pre-train on historical datasets. Whilst these are common in the real world, they are usually produced by controllers that are not optimising for data heterogeneity [16], making them smaller and less diverse than current zero-shot RL methods expect.

Can we still perform zero-shot RL using these datasets? This is the primary question this paper seeks to answer, and one we address in four parts. First, we investigate the performance of existing methods when trained on such datasets, finding their performance suffers because of out-of-distribution state-action value overestimation, a well-observed phenomenon in single-task offline RL. Second, we develop ideas from *conservatism* in single-task offline RL for use in the zero-shot RL setting, introducing a straightforward regularizer of OOD *values* or *measures* that can be used by any zero-shot RL algorithm (Figure 1). Third, we conduct experiments across varied domains, tasks and datasets, showing our *conservative* zero-shot RL proposals outperform their non-conservative counterparts, and surpass the performance of methods that get to see the task in advance. Finally, we establish

38th Conference on Neural Information Processing Systems (NeurIPS 2024).

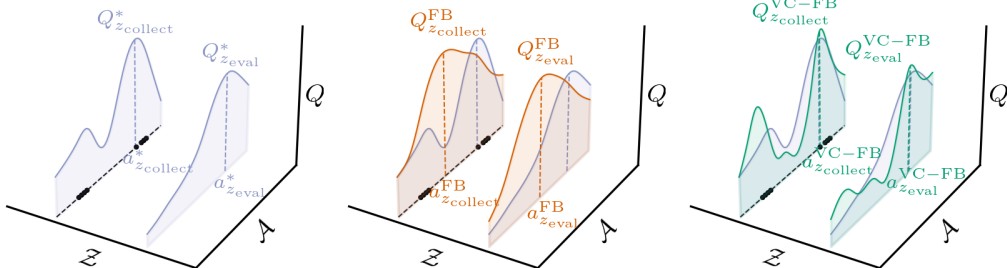

Figure 1: ***Conservative*** **zero-shot RL..** (*Left*) Zero-shot RL methods must train on a dataset collected by a behaviour policy optimising against task $z_{\text{collect}}$, yet generalise to new tasks $z_{\text{eval}}$. Both tasks have associated optimal value functions $Q^*_{z_{\text{collect}}}$ and $Q^*_{z_{\text{eval}}}$ for a given marginal state. (*Middle*) Existing methods, in this case forward-backward representations (FB), overestimate the value of actions not in the dataset for all tasks. (*Right*) Value-conservative forward-backward representations (VC-FB) suppress the value of actions not in the dataset for all tasks. Black dots (●) represent state-action samples present in the dataset.

that our proposals do not hinder performance on large heterogeneous datasets, meaning adopting them presents little downside. We believe the ideas explored in this paper represent a step toward real-world deployment of zero-shot RL methods.

## 2  Preliminaries

**Markov decision processes**  A *reward-free* Markov Decision Process (MDP) is defined by $\mathcal{M} = \{\mathcal{S}, \mathcal{A}, \mathcal{T}, \gamma\}$ where $\mathcal{S}$ is the state space, $\mathcal{A}$ is the action space, $\mathcal{T} : \mathcal{S} \times \mathcal{A} \rightarrow \Delta(\mathcal{S})$ is the transition function, where $\Delta(X)$ denotes the set of possible distributions over $X$, and $\gamma \in [0, 1)$ is the discount factor [77]. Given $(s_0, a_0) \in \mathcal{S} \times \mathcal{A}$ and a policy $\pi : \mathcal{S} \rightarrow \Delta(\mathcal{A})$, we denote $\Pr(\cdot|s_0, a_0, \pi)$ and $\mathbb{E}[\cdot|s_0, a_0, \pi]$ the probabilities and expectations under state-action sequences $(s_t, a_t)_{t \geq 0}$ starting at $(s_0, a_0)$ and following policy $\pi$, with $s_t \sim \mathcal{T}(\cdot|s_{t-1}, a_{t-1})$ and $a_t \sim \pi(\cdot|s_t)$. Given a reward function $r : \mathcal{S} \rightarrow \mathbb{R}_{\geq 0}$, the $Q$ function of $\pi$ for $r$ is $Q^\pi_r := \sum_{t \geq 0} \gamma^t \mathbb{E}[r(s_{t+1})|s_0, a_0, \pi]$.

**Zero-shot RL**  For pre-training, the agent has access to a static offline dataset of reward-free transitions $\mathcal{D} = \{(s_i, a_i, s_{i+1})\}_{i=1}^{|\mathcal{D}|}$ generated by an unknown behaviour policy, and cannot interact with the environment. At test time, a reward function $r_{\text{eval}}$ specifying a *task* is revealed and the agent must return a policy for the task without any further planning or learning. Ideally, the policy should maximise the expected discounted return on the task $\mathbb{E}[\sum_{t \geq 0} \gamma^t r_{\text{eval}}(s_{t+1})|s_0, a_0, \pi]$. The reward function is specified either via a small dataset of reward-labelled states $\mathcal{D}_{\text{labelled}} = \{(s_i, r_{\text{eval}}(s_i))\}_{i=1}^k$ with $k \leq 10,000$ or as an explicit function $s \mapsto r_{\text{eval}}(s)$ (like 1 at a goal state and 0 elsewhere). Intuitively, the zero-shot RL problem asks: is it possible to train an agent using a pre-collected dataset of transitions from an environment such that, at test time, it can return the optimal policy for any task in that environment without any further planning or learning?

State-of-the-art zero-shot RL methods leverage either successor measures [6] or successor features [5], with the former instantiated by *forward backward representations* [82] and the latter by *universal successor features* [7]. The remainder of this section introduces these ideas.

**Successor measures**  The *successor measure* $M^\pi(s_0, a_0, \cdot)$ over $\mathcal{S}$ is the cumulative discounted time spent in each future state $s_{t+1}$ after starting in state $s_0$, taking action $a_0$, and following policy $\pi$ thereafter:

$$M^\pi(s_0, a_0, X) := \sum_{t \geq 0} \gamma^t \Pr(s_{t+1} \in X|s_0, a_0, \pi) \; \forall \, X \subset \mathcal{S}. \tag{1}$$

The $Q$ function of policy $\pi$ for task $r$ is the integral of $r$ with respect to $M^\pi$:

$$Q^\pi_r(s_0, a_0) := \int_{s_+ \in \mathcal{S}} r(s_+) M^\pi(s_0, a_0, s_+). \tag{2}$$

**The forward-backward framework**  FB representations [82] approximate the successor measures of near-optimal policies for any task. Let $\rho$ be an arbitrary state distribution, and $\mathbb{R}^d$ be a representation

space. FB representations are composed of a *forward* model $F : \mathcal{S} \times \mathcal{A} \times \mathbb{R}^d \to \mathbb{R}^d$, a *backward* model $B : \mathcal{S} \to \mathbb{R}^d$, and set of polices $(\pi_z)_{z \in \mathbb{R}^d}$. They are trained such that

$$M^{\pi_z}(s_0, a_0, X) \approx \int_X F(s_0, a_0, z)^\top B(s) \rho(\mathrm{d}s) \ \ \forall \ s_0 \in \mathcal{S}, a_0 \in \mathcal{A}, X \subset \mathcal{S}, z \in \mathbb{R}^d, \quad (3)$$

and

$$\pi_z(s) \approx \arg\max_a F(s, a, z)^\top z \ \ \forall \ (s, a) \in \mathcal{S} \times \mathcal{A}, z \in \mathbb{R}^d. \quad (4)$$

Intuitively, Equation 3 says that the approximated successor measure under $\pi_z$ from $(s_0, a_0)$ to $s$ is high if their respective forward and backward embeddings are similar i.e. have large dot product. By comparing Equation 2 and Equation 3, we see that an FB representation can be used to approximate the $Q$ function of $\pi_z$ with respect to any reward function $r$ as:

$$\begin{aligned} Q_r^{\pi_z}(s_0, a_0) &\approx \int_{s \in \mathcal{S}} r(s) F(s_0, a_0, z)^\top B(s) \rho(\mathrm{d}s) \\ &= F(s_0, a_0, z)^\top \mathbb{E}_{s \sim \rho}[r(s) B(s)]. \end{aligned} \quad (5)$$

Training of $F$ and $B$ is done with TD learning [71, 78] using transition data sampled from $\mathcal{D}$:

$$\mathcal{L}_{\text{FB}} = \mathbb{E}_{(s_t, a_t, s_{t+1}, s_+) \sim \mathcal{D}, z \sim \mathcal{Z}}[(F(s_t, a_t, z)^\top B(s_+) - \gamma \bar{F}(s_{t+1}, \pi_z(s_{t+1}), z)^\top \bar{B}(s_+))^2 \\ - 2F(s_t, a_t, z)^\top B(s_{t+1})], \quad (6)$$

where $s_+$ is sampled independently of $(s_t, a_t, s_{t+1})$, $\bar{F}$ and $\bar{B}$ are lagging target networks, and $\mathcal{Z}$ is a task sampling distribution. The policy is trained in an actor-critic formulation [47]. See [82] for a full derivation of the TD update, and our Appendix B.1 for practical implementation details including the specific choice of task sampling distribution $\mathcal{Z}$.

By relating Equations 4 and 5, we find $z = \mathbb{E}_{s \sim \rho}[r(s) B(s)]$ for some reward function $r$. At test time, we can use this property to perform zero-shot RL. Using $\mathcal{D}_{\text{labelled}}$, we estimate the task as $z_{\text{eval}} \approx \mathbb{E}_{s \sim \mathcal{D}_{\text{labelled}}}[r_{\text{eval}}(s) B(s)]$ and pass it as an argument to $\pi_z$. If $z_{\text{eval}}$ lies within the task sampling distribution $\mathcal{Z}$ used during pre-training, then $\pi_z(s) \approx \arg\max_a Q_{r_{\text{eval}}}^{\pi_z}(s, a)$, and hence this policy is approximately optimal for $r_{\text{eval}}$.

**(Universal) successor features** *Successor features* assume access to a basic feature map $\varphi : \mathcal{S} \mapsto \mathbb{R}^d$ that embeds states into a representation space, and are defined as the expected discounted sum of future features $\psi^\pi(s_0, a_0) := \mathbb{E}[\sum_{t \geq 0} \gamma^t \varphi(s_{t+1}) | s_0, a_0, \pi]$ [5]. They are made *universal* by conditioning their predictions on a family of policies $\pi_z$

$$\psi(s_0, a_0, z) = \mathbb{E}\left[ \sum_{t \geq 0} \gamma^t \varphi(s_{t+1}) | s_0, a_0, \pi_z \right] \ \ \forall \ s_0 \in \mathcal{S}, a_0 \in \mathcal{A}, z \in \mathbb{R}^d, \quad (7)$$

with

$$\pi_z(s) \approx \arg\max_a \psi(s, a, z)^\top z, \ \forall \ (s_0, a_0) \in \mathcal{S} \times \mathcal{A}, z \in \mathbb{R}^d. \quad (8)$$

Like FB, USFs are trained using TD learning on

$$\mathcal{L}_{\text{SF}} = \mathbb{E}_{(s_t, a_t, s_{t+1}) \sim \mathcal{D}, z \sim \mathcal{Z}}[(\psi(s_t, a_t, z)^\top z - \varphi(s_{t+1})^\top z - \gamma \bar{\psi}(s_{t+1}, \pi_z(s_{t+1}), z)^\top z)^2], \quad (9)$$

where $\bar{\psi}$ is a lagging target network, and $\mathcal{Z}$ is the same $z$ sampling distribution used for FB. We refer the reader to [7] for a derivation of the TD update and full learning procedure. Test time policy inference is performed similarly to FB. Using $\mathcal{D}_{\text{labelled}}$, the task is inferred by performing a linear regression of $r_{\text{eval}}$ onto the features: $z_{\text{eval}} := \arg\min_z \mathbb{E}_{s \sim \mathcal{D}_{\text{labelled}}}[(r_{\text{eval}}(s) - \varphi(s)^\top z)^2]$ before it is passed as an argument to the policy.

## 3 Zero-Shot RL from Low Quality Data

In this section we introduce methods for improving the performance of zero-shot RL methods on low quality datasets. In Section 3.1, we explore the failure mode of existing methods on such datasets. Then, in Section 3.2, we propose straightforward amendments to these methods that address the failure mode. Finally, in Section 3.3, we illustrate the usefulness of our proposals with a controlled example. We develop our methods within the FB framework because of its superior empirical performance [83], but our proposals are also compatible with USF. We push their derivation to Appendix D for brevity.

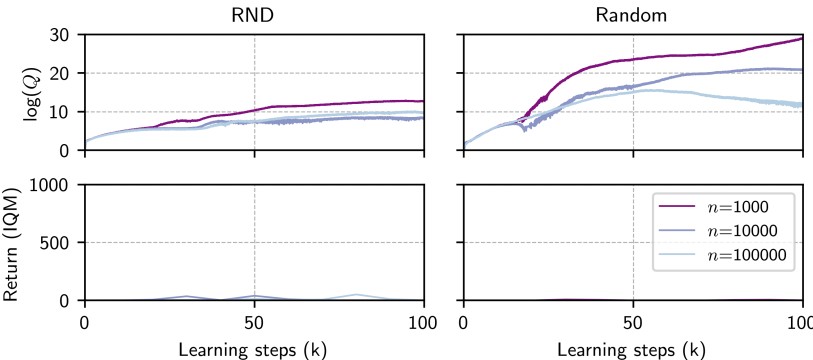

Figure 2: **FB value overestimation with respect to dataset size $n$ and quality.** Log $Q$ values and IQM of rollout performance on all Maze tasks for datasets RND and RANDOM. $Q$ values predicted during training increase as both the size and "quality" of the dataset decrease. This contradicts the low return of all resultant policies (note: a return of 1000 is the maximum achievable for this task). Informally, we say the RND dataset is "high" quality, and the RANDOM dataset is "low" quality–see Appendix A.2 for more details.

### 3.1 Failure Mode of Existing Methods

To investigate the failure mode of existing methods we examine the FB loss (Equation 6) more closely. The TD target includes an action produced by the current policy $a_{t+1} \sim \pi_z(s_{t+1})$. Equation 4 shows this is the policy's current best estimate of the $Q$-maximising action in state $s$ for task $z$. For finite datasets, this maximisation does not constrain the policy to actions observed in the dataset, and so it can become biased towards out-of-distribution (OOD) actions thought to be of high value. In such instances $F$ and $B$ are updated towards targets for which the dataset provides no support. This *distribution shift* is a well-observed phenomenon in single-task offline RL [42, 46, 44], and is exacerbated by small, low-diversity datasets as we explore in Figure 2.

### 3.2 Mitigating the Distribution Shift

In the single-task setting, the distribution shift is addressed by applying constraints to either the policy, value function or model (see Section 6 for a summary of past work). Here we re-purpose single-task value function and model regularisation for use in the zero-shot RL setting. To avoid further complicating zero-shot RL methods, we only consider regularisation techniques that do not introduce new parametric functions. We discuss the implications of this decision in Section 5.

Conservative $Q$-learning (CQL) [42, 44] regularises the $Q$ function by querying OOD state-action pairs and suppressing their value. This is achieved by adding new term to the usual $Q$ loss function

$$\mathcal{L}_{\text{CQL}} = \alpha \cdot \mathbb{E}_{s \sim \mathcal{D}, a \sim \mu(a|s)}[Q(s,a)] - \mathbb{E}_{(s,a) \sim \mathcal{D}}[Q(s,a)] - \mathcal{H}(\mu) + \mathcal{L}_{\text{Q}}, \tag{10}$$

where $\alpha$ is a scaling parameter, $\mu(a|s)$ is a policy distribution selected to find the maximum value of the current $Q$ function iterate, $\mathcal{H}(\mu)$ is the entropy of $\mu$ used for regularisation, and $\mathcal{L}_{\text{Q}}$ is the normal TD loss on $Q$. Equation 10 has the dual effect of minimising the peaks in $Q$ under $\mu$ whilst maximising $Q$ for state-action pairs in the dataset.

We can replicate a similar form of regularisation in the FB framework, substituting $F(s,a,z)^\top z$ for $Q$ in Equation 10 and adding the normal FB loss (Equation 6)

$$\mathcal{L}_{\text{VC-FB}} = \alpha \cdot (\mathbb{E}_{s \sim \mathcal{D}, a \sim \mu(a|s), z \sim \mathcal{Z}}[F(s,a,z)^\top z] - \mathbb{E}_{(s,a) \sim \mathcal{D}, z \sim \mathcal{Z}}[F(s,a,z)^\top z] - \mathcal{H}(\mu)) + \mathcal{L}_{\text{FB}}. \tag{11}$$

The key difference between Equations 10 and 11 is that the former suppresses the value of OOD actions for one task, whereas the latter does so for all task vectors drawn from $\mathcal{Z}$. We call models learnt with this loss *value-conservative forward-backward representations* (VC-FB).

Because FB derives $Q$ functions from successor measures (Equation 5), and because (by assumption) rewards are non-negative, suppressing the predicted measures for OOD actions provides an alternative route to suppressing their $Q$ values. As we did with VC-FB, we can substitute FB's successor measure approximation $F(s,a,z)^\top B(s_+)$ into Equation 10, which yields:

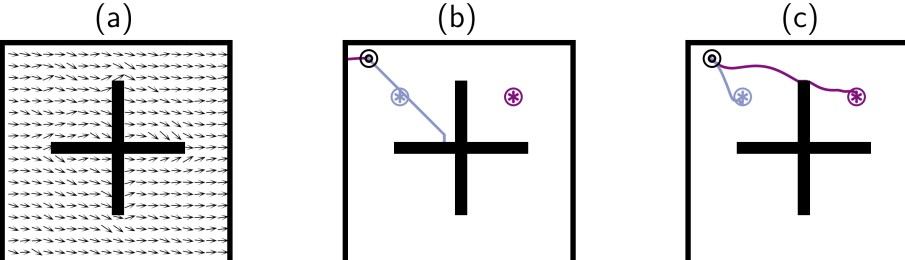

Figure 3: **Ignoring out-of-distribution actions.** The agents are tasked with learning separate policies for reaching ⊛ and ⊛. (a) RND dataset with all "left" actions removed; quivers represent the mean action direction in each state bin. (b) Best FB rollout after 1 million learning steps. (c) Best VC-FB performance after 1 million learning steps. FB overestimates the value of OOD actions and cannot complete either task; VC-FB synthesises the requisite information from the dataset and completes both tasks.

$$\mathcal{L}_{\text{MC-FB}} = \alpha \cdot (\mathbb{E}_{s \sim \mathcal{D}, a \sim \mu(a|s), z \sim \mathcal{Z}, s_+ \sim \mathcal{D}}[F(s, a, z)^\top B(s_+)]$$
$$-\mathbb{E}_{(s,a) \sim \mathcal{D}, z \sim \mathcal{Z}, s_+ \sim \mathcal{D}}[F(s, a, z)^\top B(s_+)] - \mathcal{H}(\mu)) + \mathcal{L}_{\text{FB}}. \quad (12)$$

Equation 12 has the effect of suppressing the expected visitation count to goal state $s_+$ when taking an OOD action for all task vectors drawn from $\mathcal{Z}$, which says, informally, if we don't know where OOD actions take us in the MDP, we assume they have low probability of taking us to any future states for all tasks. This is analogous to works that regularise model predictions in the single-task offline RL setting [37, 96, 69]. As such, we call this variant a *measure-conservative forward-backward representation* (MC-FB). Since it is not obvious *a priori* whether the VC-FB or MC-FB form of conservatism would be more effective in practice, we evaluate both in Section 4.

Implementing these proposals requires two new model components: 1) a conservative penalty scaling factor $\alpha$ and 2) a way of obtaining policy distribution $\mu(a|s)$ that maximises the current value or measure iterate. For 1), we observe fixed values of $\alpha$ leading to fragile performance, so dynamically tune it at each learning–see Appendix B.1.4. For 2), the choice of maximum entropy regularisation following [44]'s CQL($\mathcal{H}$) allows $\mu$ to be approximated conveniently with a log-sum exponential across $Q$ values derived from the current policy distribution and a uniform distribution. That this is true is not obvious, so we refer the reader to the detail and derivations in Section 3.2, Appendix A, and Appendix E of [44], as well as our adjustments to [44]'s theory in Appendix B.1.3. Code snippets demonstrating the required changes to a vanilla FB implementation are provided in Appendix G. We emphasise these additions represent only a small increase in the number of lines required to implement existing methods.

### 3.3 A Didactic Example

To understand situations in which a conservative zero-shot RL methods may be useful, we introduce a modified version of Maze from the ExORL benchmark [95]. Episodes begin with a point-mass initialised in the upper left of the maze (⊙), and the agent is tasked with selecting $x$ and $y$ tilt directions such that the mass is moved towards one of two goal locations (⊛ and ⊛). The action space is two-dimensional and bounded in $[-1, 1]$. We take the RND dataset and remove all "left" actions such that $a_x \in [0, 1]$ and $a_y \in [-1, 1]$, creating a dataset that has the necessary information for solving the tasks, but is inexhaustive (Figure 3 (a)). We train FB and VC-FB on this dataset and plot the highest-reward trajectories–Figure 3 (b) and (c). FB overestimates the value of OOD actions and cannot complete either task. Conversely, VC-FB synthesises the requisite information from the dataset and completes both tasks.

## 4 Experiments

In this section we perform an empirical study to evaluate our proposals. We seek answers to four questions: **(Q1)** Can our proposals from Section 3 improve FB performance on small and/or low-quality exploratory datasets? **(Q2)** How does the performance of VC-FB and MC-FB vary with respect to task type and dataset diversity? **(Q3)** Do we sacrifice performance on full datasets for

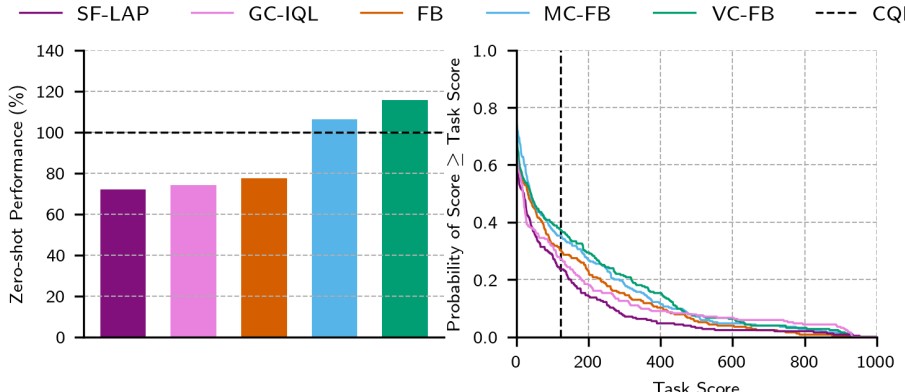

Figure 4: **Aggregate zero-shot performance on ExORL.** *(Left)* IQM of task scores across datasets and domains, normalised against the performance of CQL, our baseline. *(Right)* Performance profiles showing the distribution of scores across all tasks and domains. Both conservative FB variants stochastically dominate vanilla FB–see [1] for performance profile exposition. The black dashed line represents the IQM of CQL performance across all datasets, domains, tasks and seeds.

performance on small and/or low-quality datasets? **(Q4)** If our pre-training dataset only covers behaviour related to one downstream task (i.e. the dataset distribution is narrow and not exploratory), can our proposals from Section 3 improve FB performance on that task?

## 4.1 Setup

**Domains** We respond to **Q1-Q3** using the ExORL benchmark [95]. ExORL provides datasets collected by unsupervised exploratory algorithms on the DeepMind Control Suite [80]. We select three of the same domains as [83]: Walker, Quadruped and Maze, but substitute Jaco for Cheetah. This provides two locomotion domains and two goal-reaching domains. Within each domain, we evaluate on all tasks provided by the DeepMind Control Suite for a total of 17 tasks across four domains. Full details are provided in Appendix A.1. We respond to **Q4** using the D4RL benchmark [21]. We select the two MuJoCo [81] environments from the Open AI gym [8] that closest resemble those from ExORL: Walker2D and HalfCheetah.

**Datasets** For **Q1-Q3** we pre-train on three datasets of varying quality from ExORL. There is no unambiguous metric for quantifying dataset quality, so we use the reported performance of offline TD3 on Maze for each dataset as a proxy[1]. We choose datasets collected via Random Network Distillation (RND) [10], Diversity is All You Need (DIAYN) [18], and RANDOM policies, where agents trained on RND are the most performant, on DIAYN are median performers, and on RANDOM are the least performant. As well as selecting for quality, we also select for size by uniformly sub-sampling 100,000 transitions from each dataset. For **Q4** we choose the "medium", "medium-replay", and "medium-expert" datasets from D4RL, each providing different fractions of random, medium and expert task-directed trajectories. More details on the datasets are provided in Appendix A.2.

## 4.2 Baselines

We compare our proposals to baselines from three categories: 1) zero-shot RL methods, 2) goal-conditioned RL (GCRL) methods, and 3) single-task offline RL methods. From category 1), we use the state-of-the-art successor measure based method, FB, and the state-of-the-art successor feature based method, SF with features from Laplacian eigenfunctions (SF-LAP) [83]. From category 2), we use goal-conditioned IQL (GC-IQL) [60], a state-of-the-art GCRL method that, like our proposals, regularises the value function at OOD state-actions. We condition GC-IQL on the goal state on Maze and Jaco, and on the state in $\mathcal{D}_{\text{labelled}}$ with highest reward on Walker and Quadruped in lieu of a

---

[1]We note that [75] propose metrics that describe dataset quality as a function of the behaviour policy's *exploration* and *exploitation* w.r.t. one downstream task. However, since we are interested in generalising to *any* downstream task we cannot use these proposals directly, nor can we easily re-purpose them. We acknowledge that our proxy is imperfect, and that more work is required to better understand what dataset quality means in the context of zero-shot RL.

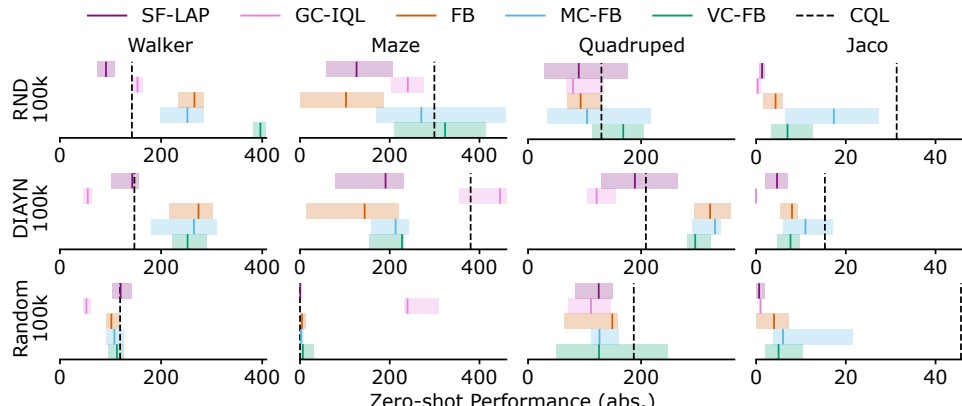

Figure 5: **Performance by dataset/domain on ExORL.** IQM scores across tasks/seeds with 95% conf. intervals.

well-defined goal state. From category 3), we use CQL and offline TD3 trained on the same datasets relabelled with task rewards. CQL approximates what an algorithm with similar mechanistics can achieve when optimising for one task in a domain rather than all tasks. Offline TD3 exhibits the best aggregate single-task performance on the ExORL benchmark, so it should be indicative of the maximum performance we could expect to extract from a dataset. Full implementation details for all algorithms are provided in Appendix B. The full evaluation protocol is described in Appendix A.5. Appendix A.6 provides a breakdown of the computational resources used in this work.

### 4.3 Results

**Q1** We report the aggregate performance of our baselines and proposals on ExORL in Figure 4. Both MC-FB and VC-FB outperform the zero-shot RL and GCRL baselines, achieving **150%** and **137%** of FB's IQM performance respectively. The performance gap between FB and SF-LAP is consistent with the results in [83]. MC-FB and VC-FB outperform our single-task baseline in expectation, reaching 111% and 120% of CQL's IQM performance

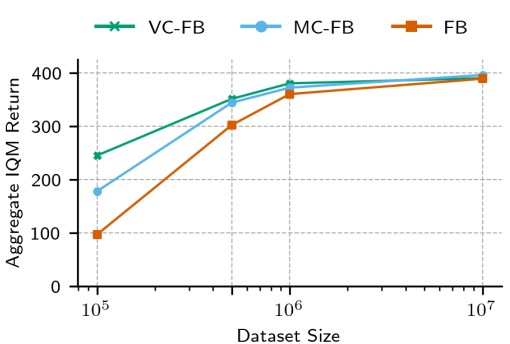

respectively *despite not having access to task-specific reward labels and needing to fit policies for all tasks*. This is a surprising result, and to the best of our knowledge, the first time a multi-task offline agent has been shown to outperform a single-task analogue. CQL outperforms offline TD3 in aggregate, so we drop offline TD3 from the core analysis, but report its full results in Appendix C alongside all other methods. We note FB achieves 80% of single-task offline TD3, which roughly aligns with the 85% performance on the full datasets reported by [83].

**Q2** We decompose the methods' performance with respect to domain and dataset diversity in Figure 5. The largest gap in performance between the conservative FB variants and FB is on RND. VC-FB and MC-FB reach 2.5× and 1.8× of FB performance respectively, and outperform CQL on three of the four domains. On DIAYN,

Figure 6: **Performance by dataset size.** Aggregate IQM scores across all domains and tasks as RND size is varied. The performance delta between vanilla FB and the conservative variants increases as dataset size decreases.

the conservative variants outperform all methods and reach 1.3× CQL's score. On the RANDOM dataset, all methods perform similarly poorly, except for CQL on Jaco, which outperforms all methods. However, in general, these results suggest the RANDOM dataset is not informative enough to extract valuable policies–discussed further in response to Q3. There appears to be little correlation between the type of domain (Appendix A.1) and the score achieved by any method. GC-IQL performs particularly well on the goal-reaching domains as expected, but worse than all zero-shot methods on the locomotion tasks, irrespective of whether they are conservative or not. This is presumably because the goal-state used to condition the policy (*i.e.* the state with highest reward in $\mathcal{D}_{\text{labelled}}$) is a poor proxy for the true, dense reward function.

Table 1: **Aggregate performance on full ExORL datasets.** IQM scores aggregated over domains and tasks for all datasets, averaged across three seeds. Both VC-FB and MC-FB maintain the performance of FB; the largest relative performance improvement is on RANDOM.

| Dataset | Domain | Task | FB | VC-FB | MC-FB |
|---|---|---|---|---|---|
| RND | all | all | 389 | 390 | 396 |
| DIAYN | all | all | 269 | 280 | 283 |
| RANDOM | all | all | 111 | 131 | 133 |
| ALL | all | all | 256 | 267 | **271** |

**Q3**  We report the aggregated performance of all FB methods across domains when trained on the full datasets in Table 1 (a full breakdown of results in provided in Appendix C). Both conservative FB variants slightly exceed the performance of vanilla FB in expectation. The largest relative performance improvement is on the RANDOM dataset–MC-FB performance is 20% higher than FB, compared to 5% higher on DIAYN and 2% higher on RND. This corroborates the hypothesis that RANDOM-100K was not informative enough to extract valuable policies.

Table 1 and Figure 4 suggest the performance gap between the conservative FB variants and vanilla FB changes as dataset size is varied. We further explore this effect in Figure 6 where we scale the RND dataset size from $10^5$ through $10^7$ and plot aggregate IQM performance of FB, VC-FB and MC-FB across all domains. We find that the performance gap decreases as dataset size increases. This result is to be expected: a larger dataset size for a fixed exploration algorithm means $a_{t+1} \sim \pi_z(s_{t+1})$ in the FB TD update (Equation 6) is more likely to be in the dataset, the policy is less likely to become biased toward OOD actions, and conservatism is less needed.

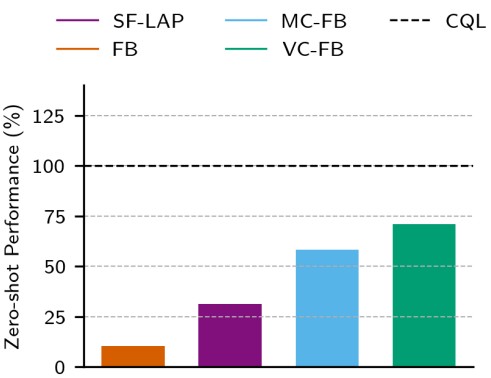

Figure 7: **Aggregate zero-shot performance on D4RL.** Aggregate IQM scores across all domains and datasets, normalised against the performance of CQL.

**Q4**  We report the aggregate performance of all zero-shot RL methods and CQL on our D4RL domains in Figure 7. FB fails all domain-dataset tasks, and reaches only 10% of CQL's aggregate performance. MC-FB and VC-FB improve on FB's considerably (by 5.6 × and 6.8 × respectively) but under-perform CQL. SF-LAP outperforms FB, but under-performs VC-FB, MC-FB and CQL.

## 5   Discussion and Limitations

**Performance discrepancy between conservative variants**  Why does VC-FB outperform MC-FB on both ExORL and D4RL? To understand, we inspect the regularising effect of both models more closely. VC-FB regularises OOD actions on $F(s, a, z)^\top z$, with $s \sim \mathcal{D}$, and $z \sim \mathcal{Z}$, whilst MC-FB regularises OOD actions on $F(s, a, z)^\top B(s_+)$, with $(s, s_+) \sim \mathcal{D}$ and $z \sim \mathcal{Z}$. Note the trailing $z$ in VC-FB is replaced with $B(s_+)$ in MC-FB which ties its updates to $\mathcal{D}$ further. We hypothesised that as $|\mathcal{D}|$ reduces, $B(s_+)$ provides poorer task coverage than $z \sim \mathcal{Z}$, hence the comparable performance on full datasets and divergent performance on 100k datasets.

To test this, we evaluate a third conservative variant called *directed* (D)VC-FB which replaces all $z \sim \mathcal{Z}$ in VC-FB with $B(s_+)$ such that OOD actions are regularised on $F(s, a, B(s_+))^\top B(s_+)$ with $(s, s_+) \sim \mathcal{D}$. This ties conservative updates entirely to $\mathcal{D}$, and according to our above hypothesis, $D$VC-FB should perform worse than VC-FB and MC-FB on the 100k ExORL datasets. See Appendix B.1.6 for implementation details. We evaluate this variant on all 100k ExORL datasets, domains and tasks and compare with FB, VC-FB and MC-FB in Table 2. See Appendix C for a full breakdown.

We find the aggregate relative performance of each method is as expected i.e. $D$VC-FB < MC-FB < VC-FB. As a consequence we conclude that VC-FB should be preferred for small datasets with

Table 2: **Aggregated performance of conservative variants employing differing $z$ sampling procedures on ExORL.** $D$VC-FB derives all $z$s from the backward model; VC-FB derives all $z$s from $\mathcal{Z}$; and MC-FB combines both. Performance correlates with the degree to which $z \sim \mathcal{Z}$.

| Dataset | Domain | Task | FB | $D$VC-FB | MC-FB | VC-FB |
|---------|--------|------|----|---------|-------|-------|
| ALL (100k) | all | all | 99 | 108 | 136 | 148 |

no prior knowledge of the dataset or test tasks. Of course, for a specific domain-dataset pair, $B(s_+)$ with $s_+ \sim \mathcal{D}$ may happen to cover the tasks well, and MC-FB may outperform VC-FB. We suspect this was the case for all datasets on the Jaco domain for example. Establishing whether this will be true *a priori* requires either relaxing the restrictions imposed by the zero-shot RL setting, or better understanding of the distribution of tasks in $z$-space and their relationship to pre-training datasets. The latter is important future work.

**Avoiding new parametric functions**   State-of-the-art zero-shot RL methods are complex, and we wanted to avoid further complicating them with new parametric functions. This limited our solution-space to CQL-style regularisation techniques, but had we relaxed this constraint, other options become available. Methods like AWAC [58], IQL [40], and $X$-QL [25] all require an estimate of the state-value function which is not immediately accessible in the FB or USF frameworks. In theory, we could learn an action-independent USF of the form $V(s, z) = \mathbb{E}[\sum_{t \geq 0} \gamma^t \varphi(s_{t+1})|s_0, \pi_z] \ \forall \ s_0 \in \mathcal{S}, z \in \mathbb{R}^d$ concurrently to $F$ and $B$ (or $\psi$ for USFs). If learnt with expectile regression, this function could be used to implement IQL and $\mathcal{X}$-QL style regularisation; without expectile regression it could be used to compute the advantage weighting required for AWAC. It's possible that implementing these methods could improve downstream performance and reduce computational overhead at the cost of increased training complexity. We leave this worthwhile investigation for future work. We provide detail of negative results related to downstream finetuning of FB models in Appendix E to help inform future research.

**D4RL Performance**   Unlike the ExORL results, VC-FB and MC-FB do not outperform CQL on the D4RL benchmark. We believe these narrower data distributions require a more careful selection of the conservative penalty scaling factor $\alpha$. We explore this further in Appendix F, and note this is corroborated by findings in the original CQL paper [44]. Methods described above, like IQL, have been shown to be more robust than CQL partly because they bypass $\alpha$ tuning. We expect that exploring the integration of these methods may improve D4RL performance.

## 6   Related Work

**Zero-shot RL**   Zero-shot RL methods build upon successor representations [15], universal value function approximators [74], successor features [5] and successor measures [6]. The state-of-the-art methods instantiate these ideas as either universal successor features (USFs) [7] or forward-backward (FB) representations [82, 83], with recent work showing the latter can be used to perform a range of imitation learning techniques efficiently [65]. A representation learning method is required to learn the features for USFs, with past works using inverse curiosity modules [62], diversity methods [49, 29], Laplacian eigenfunctions [87], or contrastive learning [13]. No works have yet explored the issues arising when training these methods on low quality offline datasets, and only one has investigated applying these ideas to real-world problems [34].

Goal-conditioned RL methods train policies to reach any goal state from any other state, and so can be used to perform zero-shot RL in goal-reaching environments [60, 54, 93, 19, 85]. However, they have no principled mechanism for conditioning policies on "dense" reward functions (as such tasks are not solved by simply reaching a particular state), and so are not full zero-shot RL methods. A concurrent line of work trains policies using sequence models conditioned on reward-labelled histories [12, 33, 45, 68, 99, 11, 24, 76, 91, 90], but, unlike zero-shot RL methods, these works do not have a robust mechanism for generalising to different reward functions as test-time.

**Offline RL**   Offline RL algorithms require regularisation of policies, value functions, models, or a combination to manage the offline-to-online distribution shift [46]. Past works regularise policies with explicit constraints [88, 20, 23, 22, 27, 64, 43, 86, 94], via important sampling [66, 79, 50, 57, 26], by leveraging uncertainty in predictions [89, 98, 4, 36], or by minimising OOD action queries [84, 14, 40], a form of imitation learning [72, 73]. Other works constrain value function approximation

so OOD action values are not overestimated [44, 42, 52, 53, 51, 92]. Offline model-based RL methods use the model to identify OOD states and penalise predicted rollouts passing through them [97, 37, 96, 2, 55, 67, 69]. All of these works have focused on regularising a finite number of policies; in contrast we extend this line of work to the zero-shot RL setting which is concerned with learning an infinite family of policies.

# 7 Conclusion

In this paper, we explored training agents to perform zero-shot reinforcement learning (RL) from low quality data. We established that the existing methods suffer in this regime because they overestimate the value of out-of-distribution state-action values, a well-observed pheneomena in single-task offline RL. As a resolution, we proposed a family of *conservative* zero-shot RL algorithms that regularise value functions or dynamics predictions on out-of-distribution state-action pairs. In experiments across various domains, tasks and datasets, we showed our proposals outperform their non-conservative counterparts in aggregate and sometimes surpass our task-specific baseline despite lacking access to reward labels *a priori*. In addition to improving performance when trained on sub-optimal datasets, we showed that performance on large, diverse datasets does not suffer as a consequence of our design decisions. Our proposals represent a step towards the use of zero-shot RL methods in the real world.

## Acknowledgements

We thank Sergey Levine for helpful feedback on the core and finetuning experiments, and Alessandro Abate and Yann Ollivier for reviewing earlier versions of this manuscript. We also thank the anonymous reviewers whose suggestions significantly improved this work. Computational resources were provided by the Cambridge Centre for Data-Driven Discovery (C2D3) and Bristol Advanced Computing Research Centre (ACRC). This work was supported by an EPSRC DTP Studentship (EP/T517847/1) and Emerson Electric.

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

# Appendices

# A Experimental Details

## A.1 ExORL Domains

We consider two locomotion and two goal-directed domains from the ExORL benchmark [95] which is built atop the DeepMind Control Suite [80]. Environments are visualised here: `https://www.youtube.com/watch?v=rAai4QzcYbs`. The domains are summarised in Table 3.

**Walker.** A two-legged robot required to perform locomotion starting from bent-kneed position. The state and action spaces are 24 and 6-dimensional respectively, consisting of joint torques, velocities and positions. ExORL provides four tasks `stand`, `walk`, `run` and `flip`. The reward function for `stand` motivates straightened legs and an upright torse; `walk` and `run` are supersets of `stand` including reward for small and large degrees of forward velocity; and `flip` motivates angular velocity of the torso after standing. Rewards are dense.

**Quadruped.** A four-legged robot required to perform locomotion inside a 3D maze. The state and action spaces are 78 and 12-dimensional respectively, consisting of joint torques, velocities and positions. ExORL provides five tasks `stand`, `roll`, `roll fast`, `jump` and `escape`. The reward function for `stand` motivates a minimum torse height and straightened legs; `roll` and `roll fast` require the robot to flip from a position on its back with varying speed; `jump` adds a term motivating vertical displacement to stand; and `escape` requires the agent to escape from a 3D maze. Rewards are dense.

**Maze.** A 2D maze with four rooms where the task is to move a point-mass to one of the rooms. The state and action spaces are 4 and 2-dimensional respectively; the state space consists of $x, y$ positions and velocities of the mass, the action space is the $x, y$ tilt angle. ExORL provides four reaching tasks `top left`, `top right`, `bottom left` and `bottom right`. The mass is always initialised in the top left and the reward is proportional to the distance from the goal, though is sparse i.e. it only registers once the agent is reasonably close to the goal.

**Jaco.** A 3D robotic arm tasked with reaching an object. The state and action spaces are 55 and 6-dimensional respectively and consist of joint torques, velocities and positions. ExORL provides four reaching tasks `top left`, `top right`, `bottom left` and `bottom right`. The reward is proportional to the distance from the goal object, though is sparse i.e. it only registers once the agent is reasonably close to the goal object.

## A.2 ExORL Datasets

We train on 100,000 transitions uniformly sampled from three datasets on the ExORL benchmark collected by different unsupervised agents: RANDOM, DIAYN, and RND. The state coverage on Maze is depicted in Figure 8. Though harder to visualise, we found that state marginals on higher-dimensional tasks (e.g. Walker) showed a similar diversity in state coverage.

**RND.** An agent whose exploration is directed by the predicted error in its ensemble of dynamics models. Informally, we say RND datasets exhibit *high* state diversity.

**DIAYN.** An agent that attempts to sequentially learn a set of skills. Informally, we say DIAYN datasets exhibit *medium* state diversity.

Table 3: **ExORL domain summary.** *Dimensionality* refers to the relative size of state and action spaces. *Type* is the task categorisation, either locomotion (satisfy a prescribed behaviour until the episode ends) or goal-reaching (achieve a specific task to terminate the episode). *Reward* is the frequency with which non-zero rewards are provided, where dense refers to non-zero rewards at every timestep and sparse refers to non-zero rewards only at positions close to the goal. Green and red colours reflect the relative difficulty of these settings.

| Domain | Dimensionality | Type | Reward |
|---|---|---|---|
| Walker | Low | Locomotion | Dense |
| Quadruped | High | Locomotion | Dense |
| Maze | Low | Goal-reaching | Sparse |
| Jaco | High | Goal-reaching | Sparse |

**RANDOM**. A agent that selects actions uniformly at random from the action space. Informally, we say RANDOM datasets exhibit *low* state diversity.

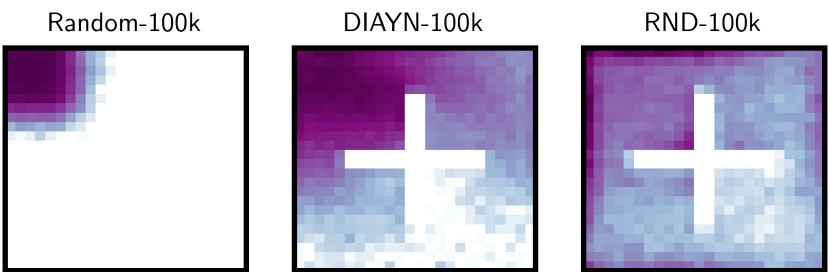

Random-100k       DIAYN-100k       RND-100k

Figure 8: **Maze state coverage by dataset.** *(left)* RANDOM; *(middle)* DIAYN; *(right)* RND.

### A.3   D4RL Domains

We consider two MuJoCo [81] locomotion tasks from the D4RL benchmark [21], which is built atop the v2 Open AI Gym [8]. The below environment descriptions are taken from [8].

**Walker2D-v2**. A two-dimensional two-legged figure that consist of seven main body parts - a single torso at the top (with the two legs splitting after the torso), two thighs in the middle below the torso, two legs in the bottom below the thighs, and two feet attached to the legs on which the entire body rests. The goal is to walk in the in the forward (right) direction by applying torques on the six hinges connecting the seven body parts.

**HalfCheetah-v2.** A 2-dimensional robot consisting of 9 body parts and 8 joints connecting them (including two paws). The goal is to apply a torque on the joints to make the cheetah run forward (right) as fast as possible, with a positive reward allocated based on the distance moved forward and a negative reward allocated for moving backward.

### A.4   D4RL Datasets

We consider three goal-directed datasets from D4RL, each providing a different proportion of expert trajectories. The below dataset descriptions are taken from [21].

**Medium.** Generated by training an SAC policy, early-stopping the training, and collecting 1M samples from this partially-trained policy.

**Medium-replay.** Generated by recording all samples in the replay buffer observed during training until the policy reaches the "medium" level of performance.

**Medium-expert.** Generated by mixing equal amounts of expert demonstrations and suboptimal data, either from a partially trained policy or by unrolling a uniform-at-random policy.

### A.5   Evaluation Protocol

We evaluate the cumulative reward (hereafter called score) achieved by VC-FB, MC-FB and our baselines on each task across five seeds. We report task scores as per the best practice recommendations of [1]. Concretely, we run each algorithm for 1 million learning steps, evaluating task scores at checkpoints of 20,000 steps. At each checkpoint, we perform 10 rollouts, record the score of each, and find the interquartile mean (IQM). We average across seeds at each checkpoint to create the learning curves reported in Appendix F. From each learning curve, we extract task scores from the learning step for which the all-task IQM is maximised across seeds. Results are reported with 95% confidence intervals obtained via stratified bootstrapping [17]. Aggregation across tasks, domains and datasets is always performed by evaluating the IQM. Full implementation details are provided in Appendix B.1.

### A.6 Computational Resources

We train our models on NVIDIA A100 GPUs. Training a single-task offline RL method to solve one task on one GPU takes approximately 4 hours. FB and SF solve one domain (for all tasks) on one GPU in approximately 4 hours. *Conservative* FB variants solve one domain (for all tasks) on one GPU in approximately 12 hours. As a result, our core experiments on the 100k datasets used approximately 110 GPU days of compute.

# B  Implementation Details

Here we detail implementations for all methods discussed in this paper. The code required to reproduce our experiments is available via https://github.com/enjeeneer/zero-shot-rl.

## B.1  Forward-Backward Representations

### B.1.1  Architecture

The forward-backward architecture described below follows the implementation by [83] exactly, other than the batch size which we reduce from 1024 to 512. We did this to reduce the computational expense of each run without limiting performance. The hyperparameter study in Appendix J of [83] shows this choice is unlikely to affect FB performance. All other hyperparameters are reported in Table 4.

**Forward Representation** $F(s, a, z)$. The input to the forward representation $F$ is always preprocessed. State-action pairs $(s, a)$ and state-task pairs $(s, z)$ have their own preprocessors which are feedforward MLPs that embed their inputs into a 512-dimensional space. These embeddings are concatenated and passed through a third feedforward MLP $F$ which outputs a $d$-dimensional embedding vector. Note: the forward representation $F$ is identical to $\psi$ used by USF so their implementations are identical (see Table 4).

**Backward Representation** $B(s)$. The backward representation $B$ is a feedforward MLP that takes a state as input and outputs a $d$-dimensional embedding vector.

**Actor** $\pi(s, z)$. Like the forward representation, the inputs to the policy network are similarly preprocessed. State-action pairs $(s, a)$ and state-task pairs $(s, z)$ have their own preprocessors which feedforward MLPs that embed their inputs into a 512-dimensional space. These embeddings are concatenated and passed through a third feedforward MLP which outputs a $a$-dimensional vector, where $a$ is the action-space dimensionality. A `Tanh` activation is used on the last layer to normalise their scale. As per [23]'s recommendations, the policy is smoothed by adding Gaussian noise $\sigma$ to the actions during training. Note the actors used by FB and USFs are identical (see Table 4).

**Misc.** Layer normalisation [3] and `Tanh` activations are used in the first layer of all MLPs to standardise the inputs.

### B.1.2  Task Sampling Distribution $\mathcal{Z}$

FB representations require a method for sampling the task vector $z$ at each learning step. [83] employ a mix of two methods, which we replicate:

1. Uniform sampling of $z$ on the hypersphere surface of radius $\sqrt{d}$ around the origin of $\mathbb{R}^d$,

2. Biased sampling of $z$ by passing states $s \sim \mathcal{D}$ through the backward representation $z = B(s)$. This also yields vectors on the hypersphere surface due to the $L2$ normalisation described above, but the distribution is non-uniform.

We sample $z$ 50:50 from these methods at each learning step.

### B.1.3  Maximum Value Approximator $\mu$

The conservative variants of FB require access to a policy distribution $\mu(a|s)$ that maximises the value of the current $Q$ iterate in expectation. Recall the standard CQL loss

Table 4: **Hyperparameters for zero-shot RL methods.** The additional hyperparameters for Conservative FB representations are highlighted in blue.

| Hyperparameter | Value |
|---|---|
| Latent dimension $d$ | 50 (100 for maze) |
| $F$ / $\psi$ dimensions | (1024, 1024) |
| $B$ / $\varphi$ dimensions | (256, 256, 256) |
| Preprocessor dimensions | (1024, 1024) |
| Std. deviation for policy smoothing $\sigma$ | 0.2 |
| Truncation level for policy smoothing | 0.3 |
| Learning steps | 1,000,000 |
| Batch size | 512 |
| Optimiser | Adam [38] |
| Learning rate | 0.0001 |
| Discount $\gamma$ | 0.98 (0.99 for maze) |
| Activations (unless otherwise stated) | ReLU |
| Target network Polyak smoothing coefficient | 0.01 |
| $z$-inference labels | 10,000 |
| $z$ mixing ratio | 0.5 |
| Conservative budget $\tau$ | 50 (45 for D4RL) |
| OOD action samples per policy $N$ | 3 |

$$\mathcal{L}_{\mathrm{CQL}} = \alpha \cdot \big(\mathbb{E}_{s\sim\mathcal{D}, a\sim\mu(a|s)}[Q(s,a)] - \mathbb{E}_{(s,a)\sim\mathcal{D}}[Q(s,a)] - R(\mu)\big) + \mathcal{L}_{\mathrm{Q}}, \tag{13}$$

where $\alpha$ is a scaling parameter, $\mu(a|s)$ the policy distribution we seek, $R$ regularises $\mu$ and $\mathcal{L}_{\mathrm{Q}}$ represents the normal TD loss on $Q$. [44]'s most performant CQL variant (CQL($\mathcal{H}$)) utilises maximum entropy regularisation on $\mu$ i.e. $R = \mathcal{H}(\mu)$. They show that obtaining $\mu$ can be cast as a closed-form optimisation problem of the form:

$$\max_{\mu} \mathbb{E}_{x\sim\mu(x)}[f(x)] + \mathcal{H}(\mu) \text{ s.t.} \sum_{x} \mu(x) = 1, \mu(x) \geq 0 \ \forall x, \tag{14}$$

and has optimal solution $\mu^*(x) = \frac{1}{Z}\exp(f(x))$, where $Z$ is a normalising factor. Plugging Equation 14 into Equation 13 we obtain:

$$\mathcal{L}_{\mathrm{CQL}} = \alpha \cdot \left(\mathbb{E}_{s\sim\mathcal{D}}[\log\sum_{a}\exp(Q(s,a))] - \mathbb{E}_{(s,a)\sim\mathcal{D}}[Q(s,a)]\right) + \mathcal{L}_{\mathrm{Q}}. \tag{15}$$

In discrete action spaces the `logsumexp` can be computed exactly; in continuous action spaces [44] approximate it via importance sampling using actions sampled uniformly at random, actions from the current policy conditioned on $s_t \sim \mathcal{D}$, and from the current policy conditioned on $s_{t+1} \sim \mathcal{D}^2$:

---

[2]Conditioning on next states $s_{t+1} \sim \mathcal{D}$ is not mentioned in the paper, but is present in their official implementation.

$$\log \sum_a \exp Q(s_t, a_t) = \log(\frac{1}{3} \sum_a \exp Q(s_t, a_t)) + \frac{1}{3} \sum_a \exp Q(s_t, a_t)) + \frac{1}{3} \sum_a \exp(\exp Q(s_t, a_t)),$$

$$= \log(\frac{1}{3} \mathbb{E}_{a_t \sim \text{Unif}(\mathcal{A})} \left[ \frac{\exp(Q(s_t, a_t)}{\text{Unif}(\mathcal{A})} \right] + \frac{1}{3} \mathbb{E}_{a_t \sim \pi(a_t|s_t)} \left[ \frac{\exp(Q(s_t, a_t))}{\pi(a_t|s_t)} \right]$$

$$\frac{1}{3} \mathbb{E}_{a_{t+1} \sim \pi(a_{t+1}|s_{t+1})} \left[ \frac{\exp(Q(s_t, a_t))}{\pi(a_{t+1}|s_{t+1})} \right]),$$

$$= \log(\frac{1}{3N} \sum_{a_i \sim \text{Unif}(\mathcal{A})}^{N} \left[ \frac{\exp(Q(s_t, a_t))}{\text{Unif}(\mathcal{A})} \right] + \frac{1}{6N} \sum_{a_i \sim \pi(a_t|s_t)}^{2N} \left[ \frac{\exp(Q(s_t, a_t))}{\pi(a_i|s_t)} \right]$$

$$\frac{1}{3N} \sum_{a_i \sim \pi(a_{t+1}|s_{t+1})}^{N} \left[ \frac{\exp(Q(s_t, a_t))}{\pi(a_i|s_{t+1})} \right]),$$

$$(16)$$

with $N$ a hyperparameter defining the number of actions to sample across the action-space. We can substitute $F(s, a, z)^\top z$ for $Q(s, a)$ in the final expression of Equation 17 to obtain the equivalent for VC-FB:

$$\log \sum_a \exp F(s_t, a_i, z)^\top z = \log(\frac{1}{3N} \sum_{a_i \sim \text{Unif}(\mathcal{A})}^{N} \left[ \frac{\exp(F(s_t, a_i, z)^\top z)}{\text{Unif}(\mathcal{A})} \right] + \frac{1}{6N} \sum_{a_i \sim \pi(a_t|s_t)}^{2N} \left[ \frac{\exp(F(s_t, a_i, z^\top z)}{\pi(a_i|s_t)} \right]$$

$$\frac{1}{3N} \sum_{a_i \sim \pi(a_{t+1}|s_{t+1})}^{N} \left[ \frac{\exp(F(s_t, a_i, z)^\top z)}{\pi(a_i|s_{t+1})} \right]).$$

$$(17)$$

In Appendix F, Figure 16 we show how the performance of VC-FB varies with the number of action samples. In general, performance improves with the number of action samples, but we limit $N = 3$ to limit computational burden. The formulation for MC-FB is identical other than each value $F(s, a, z)^T z$ being replaced with measures $F(s, a, z)^T B(s_+)$.

### B.1.4 Dynamically Tuning $\alpha$

A critical hyperparameter is $\alpha$ which weights the conservative penalty with respect to other losses during each update. We initially trialled constant values of $\alpha$, but found performance to be fragile to this selection, and lacking robustness across environments. Instead, we follow [44] once again, and instantiate their algorithm for dynamically tuning $\alpha$, which they call Lagrangian dual gradient-descent on $\alpha$. We introduce a conservative budget parameterised by $\tau$, and set $\alpha$ with respect to this budget:

$$\min_{FB} \max_{\alpha \geq 0} \alpha \cdot \left( \mathbb{E}_{s \sim \mathcal{D}, a \sim \mu(a|s) z \sim \mathcal{Z}}[F(s, a, z)^\top z] - \mathbb{E}_{(s,a) \sim \mathcal{D}, z \sim \mathcal{Z}}[F(s, a, z)^\top z] - \tau \right) + \mathcal{L}_{\text{FB}}. \quad (18)$$

Intuitively, this implies that if the scale of overestimation $\leq \tau$ then $\alpha$ is set close to 0, and the conservative penalty does not affect the updates. If the scale of overestimation $\geq \tau$ then $\alpha$ is set proportionally to this gap, and thus the conservative penalty is proportional to the degree of overestimation above $\tau$. As above, for the MC-FB variant values $F(s, a, z)^\top z$ are replaced with measures $F(s, a, z)^\top B(s_+)$.

### B.1.5 Algorithm

We summarise the end-to-end implementation of VC-FB as pseudo-code in Algorithm 1. MC-FB representations are trained identically other than at line 10 where the conservative penalty is computed for $M$ instead of $Q$, and in line 12 where $M$s are lower bounded via Equation 12.

---

**Algorithm 1** Pre-training value-conservative forward-backward representations

---

**Require:** $\mathcal{D}$: dataset of trajectories
          $F_{\theta_F}, B_{\theta_B}, \pi$: randomly initialised networks
          $N, \mathcal{Z}, \nu, b$: learning steps, z-sampling distribution, polyak momentum, batch size

1: **for** learning step $n = 1...N$ **do**
2:    $\{(s_i, a_i, s_{i+1})\} \sim \mathcal{D}_{i\in|b|}$                         ◁ *Sample mini-batch of transitions*
3:    $\{z_i\}_{i\in|b|} \sim \mathcal{Z}$                                 ◁ *Sample zs (Appendix B.1.2)*
4:
5:    *// FB Update*
6:    $\{a_{i+1}\} \sim \pi(s_{i+1}, z_i)$          ◁ *Sample batch of actions at next states from policy*
7:    Update $FB$ given $\{(s_i, a_i, s_{i+1}, a_{i+1}, z_i)\}$           ◁ *Equation 6*
8:
9:    *// Conservative Update*
10:   $Q^{\max}(s_i, a_i) \approx \log \sum_a \exp F(s_i, a_i, z_i)^\top z_i$   ◁ *Compute conservative penalty (Equation 17)*
11:   Compute $\alpha$ given $Q^{\max}$ via Lagrangian dual gradient-descent       ◁ *Equation 18*
12:   Lower bound $Q$                                       ◁ *Equation 11*
13:
14:   *// Actor Update*
15:   $\{a_i\} \sim \pi(s_i, z_i)$                             ◁ *Sample actions from policy*
16:   Update actor to maximise $\mathbb{E}[F(s_i, a_i, z_i)^\top z_i]$      ◁ *Standard actor-critic formulation*

17:
18:   *// Update target networks via polyak averaging*
19:   $\theta_F^- \leftarrow \nu\theta_F^- + (1-\nu)\theta_F$                   ◁ *Forward target network*
20:   $\theta_B^- \leftarrow \nu\theta_B^- + (1-\nu)\theta_B$                  ◁ *Backward target network*
21: **end for**

---

### B.1.6 *Directed* Value-Conservative Forward Backward Representations

VC-FB applies conservative updates using task vectors $z$ sampled from $\mathcal{Z}$ (which in practice is a uniform distribution over the $\sqrt{d}$-hypersphere). This will include many vectors corresponding to tasks that are never evaluated in practice in downstream applications. Intuitively, it may seem reasonable to direct conservative updates to focus on tasks that are likely to be encountered downstream. One simple way of doing this would be consider the set of all goal-reaching tasks for goal states in the training distribution, which corresponds to sampling $z = B(s_g)$ for some $s_g \sim \mathcal{D}$. This leads to the following conservative loss function:

$$\mathcal{L}_{DVC\text{-}FB} = \alpha \cdot \Big(\mathbb{E}_{s\sim\mathcal{D}, a\sim\mu(a|s), s_g\sim\mathcal{D}}[F(s, a, B(s_g))^\top B(s_g)]$$
$$- \mathbb{E}_{(s,a)\sim\mathcal{D}, s_g\sim\mathcal{D}}[F(s, a, B(s_g))^\top B(s_g)] - \mathcal{H}(\mu)\Big) + \mathcal{L}_{FB}. \quad (19)$$

We call models learnt via this loss *directed*-VC-FB (*D*VC-FB). While we were initially open to the possibility that *D*VC-FB updates would be better targeted than those of VC-FB, and would lead to improved downstream task performance, this turns out not to be the case in our experimental settings as discussed in Section 5. We report scores obtained by the *D*VC-FB method across all 100k datasets, domains and tasks in Appendix C.

## B.2 Universal Successor Features

We directly reimplement USFs, with basic features $\varphi(s)$ provided by Laplacian eigenfunctions [87], from [83].

### B.2.1 Architecture

**USF** $\psi(s, a, z)$. The input to the USF $\psi$ is always preprocessed. State-action pairs $(s, a)$ and state-task pairs $(s, z)$ have their own preprocessors which are feedforward MLPs that embed their inputs into a 512-dimensional space. These embeddings are concatenated and passed through a third

feedforward MLP $\psi$ which outputs a $d$-dimensional embedding vector. Note this is identical to the implementation of $F$ as described in Appendix B.1. All other hyperparameters are reported in Table 4.

**Feature Embedding** $\varphi(s)$**.** The feature map $\varphi(s)$ is a feedforward MLP that takes a state as input and outputs a $d$-dimensional embedding vector. The loss function for learning the feature embedding is provided in Appendix B.2.2.

**Actor** $\pi(s, z)$**.** Like the forward representation, the inputs to the policy network are similarly preprocessed. State-action pairs $(s, a)$ and state-task pairs $(s, z)$ have their own preprocessors which are feedforward MLPs that embed their inputs into a 512-dimensional space. These embeddings are concatenated and passed through a third feedforward MLP which outputs a $a$-dimensional vector, where $a$ is the action-space dimensionality. A Tanh activation is used on the last layer to normalise their scale. As per [23]'s recommendations, the policy is smoothed by adding Gaussian noise $\sigma$ to the actions during training. Note this is identical to the implementation of $\pi(s, z)$ as described in Appendix B.1.

**Misc.** Layer normalisation [3] and Tanh activations are used in the first layer of all MLPs to standardise the inputs. $z$ sampling distribution $\mathcal{Z}$ is identical to FB's (Appendix B.1.2).

### B.2.2  Laplacian Eigenfunctions Loss

Laplacian eigenfunction features $\varphi(s)$ are learned as per [87]. They consider the symmetrized MDP graph Laplacian created by some policy $\pi$, defined as $L = \mathrm{Id} - \frac{1}{2}(\mathcal{P}_\pi \mathrm{diag}\rho^-1 + \mathrm{diag}\rho^-1(\mathcal{P}_\pi)^T)$. They learn the eigenfunctions of $L$ with the following:

$$\min_{\varphi} \mathbb{E}_{(s_t, s_{t+1}) \sim \mathcal{D}} \left[ ||\varphi(s_t) - \varphi(s_{t+1})||^2 \right] + \lambda \mathbb{E}_{(s, s_+) \sim \mathcal{D}} \left[ (\varphi(s)^\top \varphi(s_+))^2 - ||\varphi(s)||_2^2 - ||\varphi(s_+)||_2^2 \right],$$

(20)

which comes from [39].

## B.3  GC-IQL

### B.3.1  Architecture

We implement GC-IQL following [60]'s codebase. GC-IQL inherits all functionality from a base soft actor-critic agent [28], but adds a soft conservative penalty to the goal-conditioned critic's $V(s, g)$ updates. We refer the reader to paper that introduces GC-IQL [60] for details on the loss function used to train $V(s, g)$. Hyperparameters are reported in Table 5.

**Critic(s).** GC-IQL trains double goal-conditioned value functions $V(s, g)$. The critics are feedforward MLPs that take a state-goal pair $(s, g)$ as input and output a value $\in \mathbb{R}^1$. During training the goals are sampled from the prior $\mathcal{G}$ described in Section B.3.2.

**Actor.** The actor is a standard feedforward MLP taking the state $s$ as input and outputting an $2a$-dimensional vector, where $a$ is the action-space dimensionality. The actor predicts the mean and standard deviation of a Gaussian distribution for each action dimension; during training a value is sampled at random, during evaluation the mean is used.

### B.3.2  Goal Sampling Distribution $\mathcal{G}$

Following [60], goals are sampled from either random states, future states, or the current state with probabilities $0.3, 0.5$ and $0.2$ respectively. A geometric distribution $\mathrm{Geom}(1 - \gamma)$ is used for the future state distribution, and the uniform distribution over the offline dataset is used for sampling random states.

## B.4  CQL

### B.4.1  Architecture

We adopt the same implementation and hyperparameters as is used on the ExORL benchmark. CQL inherits all functionality from a base soft actor-critic agent [28], but adds a conservative penalty to the critic updates (Equation 10). Hyperparameters are reported in Table 5.

**Critic(s).** CQL employs double Q networks, where the target network is updated with Polyak averaging via a momentum coefficient. The critics are feedforward MLPs that take a state-action pair $(s, a)$ as input and output a value $\in \mathbb{R}^1$.

**Actor.** The actor is a standard feedforward MLP taking the state $s$ as input and outputting an $2a$-dimensional vector, where $a$ is the action-space dimensionality. The actor predicts the mean and standard deviation of a Gaussian distribution for each action dimension; during training a value is sampled at random, during evaluation the mean is used.

### B.5   TD3

#### B.5.1   Architecture

We adopt the same implementation and hyperparameters as is used on the ExORL benchmark. Hyperparameters are reported in Table 5.

**Critic(s).** TD3 employs double Q networks, where the target network is updated with Polyak averaging via a momentum coefficient. The critics are feedforward MLPs that take a state-action pair $(s, a)$ as input and output a value $\in \mathbb{R}^1$.

**Actor.** The actor is a standard feedforward MLP taking the state $s$ as input and outputting an $a$-dimensional vector, where $a$ is the action-space dimensionality. The policy is smoothed by adding Gaussian noise $\sigma$ to the actions during training.

**Misc.** As is usual with TD3, layer normalisation [3] is applied to the inputs of all networks.

Table 5: **Hyperparameters for *Non*-zero-shot RL.**

| Hyperparameter | CQL | Offline TD3 | GC-IQL |
|---|---|---|---|
| Critic dimensions | (1024, 1024) | (1024, 1024) | (1024, 1024) |
| Actor dimensions | (1024, 1024) | (1024, 1024) | (1024, 1024) |
| Learning steps | 1,000,000 | 1,000,000 | 1,000,000 |
| Batch size | 1024 | 1024 | 1024 |
| Optimiser | Adam | Adam | Adam |
| Learning rate | 0.0001 | 0.0001 | 0.0001 |
| Discount $\gamma$ | 0.98 (0.99 for maze) | 0.98 (0.99 for maze) | 0.98 (0.99 for maze) |
| Activations | ReLU | ReLU | ReLU |
| Target network Polyak smoothing coefficient | 0.01 | 0.01 | 0.01 |
| Sampled Actions Number | 3 | - | - |
| CQL $\alpha$ | 0.01 | - | - |
| CQL Lagrange | False | - | - |
| Std. deviation for policy smoothing $\sigma$ | - | 0.2 | - |
| Truncation level for policy smoothing | - | 0.3 | - |
| IQL temperature | - | - | 1 |
| IQL Expectile | - | - | 0.7 |

### B.6   Code References

This work was enabled by: NumPy [30], PyTorch [61], Pandas [56] and Matplotlib [31].

## C   Extended Results

In this section we report a full breakdown of our experimental results on the ExORL benchmark by dataset, domain and task. Table 6 reports results for methods trained on the 100k sub-sampled datasets, and Table 7 reports results for methods trained on the full datasets.

Table 6: **100k dataset experimental results on ExORL.** For each dataset-domain pair, we report the score at the step for which the all-task IQM is maximised when averaging across 5 seeds, and the constituent task scores at that step. Bracketed numbers represent the 95% confidence interval obtained by a stratified bootstrap.

| Dataset | Domain | Task | CQL | Offline TD3 | GC-IQL | SF-LAP | FB | VC-FB (ours) | $D$VC-FB (ours) | MC-FB (ours) |
|---|---|---|---|---|---|---|---|---|---|---|
| RND-100k | Walker | Walk | $138_{(122-140)}$ | $210_{(205-231)}$ | $118_{(114-126)}$ | $58_{(33-104)}$ | $184_{(123-278)}$ | $446_{(435-460)}$ | $394_{(166-512)}$ | $247_{(164-299)}$ |
| | | Stand | $386_{(375-391)}$ | $362_{(335-378)}$ | $284_{(260-313)}$ | $190_{(128-233)}$ | $558_{(498-637)}$ | $624_{(604-639)}$ | $590_{(557-622)}$ | $480_{(402-517)}$ |
| | | Run | $71_{(64-75)}$ | $84_{(79-88)}$ | $51_{(42-56)}$ | $34_{(27-41)}$ | $101_{(90-135)}$ | $179_{(165-197)}$ | $134_{(77-191)}$ | $106_{(72-137)}$ |
| | | Flip | $153_{(135-174)}$ | $162_{(148-171)}$ | $157_{(152-173)}$ | $70_{(56-84)}$ | $163_{(90-212)}$ | $325_{(292-350)}$ | $250_{(215-286)}$ | $164_{(131-192)}$ |
| | Quadruped | Stand | $167_{(73-268)}$ | $119_{(9-338)}$ | $43_{(14-173)}$ | $108_{(51-192)}$ | $134_{(86-176)}$ | $331_{(190-410)}$ | $269_{(152-385)}$ | $171_{(71-372)}$ |
| | | Roll Fast | $93_{(27-219)}$ | $63_{(4-223)}$ | $66_{(14-92)}$ | $80_{(21-169)}$ | $83_{(55-127)}$ | $141_{(87-182)}$ | $146_{(85-207)}$ | $81_{(19-199)}$ |
| | | Roll | $251_{(147-320)}$ | $96_{(8-272)}$ | $224_{(123-399)}$ | $100_{(22-277)}$ | $139_{(71-224)}$ | $141_{(107-212)}$ | $209_{(123-295)}$ | $132_{(40-267)}$ |
| | | Jump | $128_{(82-223)}$ | $85_{(6-248)}$ | $152_{(39-247)}$ | $94_{(28-189)}$ | $121_{(78-186)}$ | $159_{(110-212)}$ | $167_{(100-234)}$ | $97_{(41-172)}$ |
| | | Escape | $3_{(2-4)}$ | $3_{(0-9)}$ | $1_{(0-3)}$ | $1_{(1-4)}$ | $7_{(3-12)}$ | $8_{(4-14)}$ | $13_{(6-19)}$ | $5_{(1-12)}$ |
| | Maze | Reach Top Right | $433_{(275-558)}$ | $457_{(0-728)}$ | $308_{(123-494)}$ | $1_{(0-368)}$ | $0_{(0-26)}$ | $0_{(0-406)}$ | $0_{(0-0)}$ | $99_{(16-432)}$ |
| | | Reach Top Left | $561_{(503-758)}$ | $921_{(897-936)}$ | $628_{(384-872)}$ | $302_{(18-602)}$ | $384_{(0-735)}$ | $662_{(218-899)}$ | $244_{(10-477)}$ | $723_{(363-895)}$ |
| | | Reach Bottom Right | $0_{(0-0)}$ | $0_{(0-0)}$ | $0_{(0-0)}$ | $0_{(0-0)}$ | $0_{(0-0)}$ | $0_{(0-0)}$ | $0_{(0-0)}$ | $0_{(0-0)}$ |
| | | Reach Bottom Left | $253_{(69-419)}$ | $85_{(22-295)}$ | $25_{(4-46)}$ | $0_{(0-34)}$ | $0_{(0-0)}$ | $479_{(56-725)}$ | $250_{(0-501)}$ | $384_{(125-653)}$ |
| | Jaco | Reach Top Right | $37_{(21-54)}$ | $0_{(0-1)}$ | $0_{(0-0)}$ | $0_{(0-1)}$ | $0_{(0-3)}$ | $1_{(0-4)}$ | $6_{(3-9)}$ | $17_{(8-29)}$ |
| | | Reach Top Left | $21_{(12-33)}$ | $0_{(0-0)}$ | $0_{(0-1)}$ | $2_{(0-5)}$ | $2_{(1-4)}$ | $2_{(1-2)}$ | $11_{(7-16)}$ | $9_{(2-21)}$ |
| | | Reach Bottom Right | $37_{(21-53)}$ | $0_{(0-0)}$ | $0_{(0-2)}$ | $0_{(0-0)}$ | $0_{(0-12)}$ | $5_{(2-21)}$ | $7_{(3-11)}$ | $16_{(7-23)}$ |
| | | Reach Bottom Left | $20_{(18-28)}$ | $0_{(0-0)}$ | $0_{(0-1)}$ | $1_{(0-4)}$ | $7_{(4-15)}$ | $4_{(1-21)}$ | $3_{(1-5)}$ | $11_{(1-41)}$ |
| RANDOM-100k | Walker | Walk | $126_{(112-139)}$ | $132_{(105-166)}$ | $24_{(21-26)}$ | $129_{(118-139)}$ | $76_{(55-116)}$ | $122_{(86-140)}$ | $38_{(32-43)}$ | $119_{(58-210)}$ |
| | | Stand | $246_{(199-287)}$ | $295_{(251-326)}$ | $141_{(115-162)}$ | $206_{(161-266)}$ | $237_{(212-278)}$ | $223_{(206-241)}$ | $223_{(201-246)}$ | $209_{(190-239)}$ |
| | | Run | $31_{(23-47)}$ | $57_{(38-65)}$ | $20_{(16-23)}$ | $49_{(38-62)}$ | $37_{(33-48)}$ | $40_{(36-46)}$ | $31_{(25-36)}$ | $32_{(27-37)}$ |
| | | Flip | $115_{(102-128)}$ | $72_{(47-83)}$ | $22_{(19-24)}$ | $100_{(82-119)}$ | $47_{(40-62)}$ | $62_{(40-99)}$ | $47_{(43-52)}$ | $44_{(40-55)}$ |
| | Quadruped | Stand | $186_{(70-294)}$ | $264_{(68-472)}$ | $76_{(16-235)}$ | $285_{(146-432)}$ | $278_{(154-440)}$ | $269_{(48-618)}$ | $196_{(100-284)}$ | $172_{(78-284)}$ |
| | | Roll Fast | $161_{(99-223)}$ | $151_{(31-283)}$ | $99_{(71-103)}$ | $64_{(26-112)}$ | $96_{(16-195)}$ | $43_{(17-132)}$ | $155_{(89-220)}$ | $78_{(53-126)}$ |
| | | Roll | $326_{(218-430)}$ | $260_{(41-463)}$ | $165_{(37-264)}$ | $111_{(66-169)}$ | $105_{(63-185)}$ | $130_{(74-185)}$ | $183_{(120-246)}$ | $178_{(115-452)}$ |
| | | Jump | $213_{(136-293)}$ | $189_{(82-380)}$ | $139_{(18-397)}$ | $128_{(14-221)}$ | $75_{(33-155)}$ | $78_{(23-226)}$ | $94_{(67-121)}$ | $147_{(53-261)}$ |
| | | Escape | $6_{(2-8)}$ | $4_{(2-9)}$ | $1_{(0-5)}$ | $2_{(0-5)}$ | $5_{(3-7)}$ | $2_{(1-11)}$ | $3_{(2-5)}$ | $6_{(1-10)}$ |
| | Maze | Reach Top Right | $0_{(0-0)}$ | $0_{(0-0)}$ | $0_{(0-0)}$ | $0_{(0-0)}$ | $0_{(0-0)}$ | $0_{(0-0)}$ | $0_{(0-0)}$ | $0_{(0-0)}$ |
| | | Reach Top Left | $0_{(0-0)}$ | $0_{(0-2)}$ | $925_{(915-929)}$ | $3_{(0-5)}$ | $18_{(0-54)}$ | $26_{(4-129)}$ | $52_{(0-104)}$ | $10_{(0-32)}$ |
| | | Reach Bottom Right | $0_{(0-0)}$ | $0_{(0-0)}$ | $0_{(0-0)}$ | $0_{(0-0)}$ | $0_{(0-0)}$ | $0_{(0-0)}$ | $0_{(0-0)}$ | $0_{(0-0)}$ |
| | | Reach Bottom Left | $0_{(0-0)}$ | $0_{(0-4)}$ | $37_{(0-233)}$ | $0_{(0-0)}$ | $0_{(0-0)}$ | $0_{(0-0)}$ | $0_{(0-0)}$ | $0_{(0-0)}$ |
| | Jaco | Reach Top Right | $53_{(47-60)}$ | $34_{(17-89)}$ | $0_{(0-0)}$ | $0_{(0-0)}$ | $3_{(0-15)}$ | $0_{(0-8)}$ | $0_{(0-0)}$ | $4_{(0-11)}$ |
| | | Reach Top Left | $52_{(34-88)}$ | $2_{(1-5)}$ | $4_{(2-5)}$ | $2_{(0-5)}$ | $0_{(0-0)}$ | $12_{(7-25)}$ | $26_{(10-42)}$ | $23_{(11-53)}$ |
| | | Reach Bottom Right | $53_{(45-60)}$ | $34_{(15-78)}$ | $0_{(0-0)}$ | $0_{(0-0)}$ | $0_{(0-4)}$ | $1_{(0-1)}$ | $30_{(0-59)}$ | $1_{(0-6)}$ |
| | | Reach Bottom Left | $32_{(19-37)}$ | $3_{(1-4)}$ | $0_{(0-0)}$ | $0_{(0-0)}$ | $2_{(1-12)}$ | $0_{(0-0)}$ | $16_{(0-33)}$ | $0_{(0-8)}$ |
| DIAYN-100k | Walker | Walk | $147_{(123-198)}$ | $150_{(132-164)}$ | $23_{(21-26)}$ | $93_{(61-106)}$ | $251_{(132-299)}$ | $262_{(174-344)}$ | $248_{(243-255)}$ | $260_{(175-347)}$ |
| | | Stand | $406_{(365-441)}$ | $262_{(234-300)}$ | $142_{(117-173)}$ | $276_{(189-292)}$ | $497_{(381-651)}$ | $455_{(401-491)}$ | $387_{(352-423)}$ | $423_{(370-594)}$ |
| | | Run | $38_{(33-42)}$ | $45_{(44-47)}$ | $19_{(15-23)}$ | $53_{(37-59)}$ | $98_{(79-114)}$ | $82_{(76-96)}$ | $87_{(82-92)}$ | $81_{(71-107)}$ |
| | | Flip | $149_{(116-178)}$ | $163_{(152-179)}$ | $23_{(20-32)}$ | $144_{(89-159)}$ | $193_{(136-205)}$ | $228_{(193-244)}$ | $180_{(155-205)}$ | $182_{(151-237)}$ |
| | Quadruped | Stand | $299_{(160-435)}$ | $848_{(722-885)}$ | $251_{(213-404)}$ | $313_{(167-492)}$ | $459_{(396-525)}$ | $430_{(393-481)}$ | $447_{(413-482)}$ | $457_{(396-511)}$ |
| | | Roll Fast | $164_{(75-195)}$ | $446_{(350-499)}$ | $111_{(84-160)}$ | $185_{(162-319)}$ | $287_{(256-328)}$ | $260_{(232-280)}$ | $290_{(285-296)}$ | $293_{(275-299)}$ |
| | | Roll | $264_{(126-369)}$ | $709_{(619-799)}$ | $117_{(41-209)}$ | $189_{(98-306)}$ | $460_{(411-485)}$ | $415_{(396-434)}$ | $429_{(407-452)}$ | $456_{(408-490)}$ |
| | | Jump | $196_{(135-267)}$ | $410_{(350-517)}$ | $171_{(141-213)}$ | $240_{(102-350)}$ | $363_{(337-418)}$ | $357_{(324-397)}$ | $391_{(371-411)}$ | $372_{(329-403)}$ |
| | | Escape | $6_{(3-11)}$ | $23_{(15-32)}$ | $6_{(3-9)}$ | $16_{(6-28)}$ | $45_{(35-56)}$ | $31_{(24-43)}$ | $45_{(42-48)}$ | $42_{(37-50)}$ |
| | Maze | Reach Top Right | $760_{(494-787)}$ | $796_{(520-799)}$ | $705_{(402-777)}$ | $0_{(0-0)}$ | $0_{(0-0)}$ | $0_{(0-0)}$ | $0_{(0-0)}$ | $27_{(0-97)}$ |
| | | Reach Top Left | $943_{(941-950)}$ | $943_{(941-945)}$ | $901_{(889-915)}$ | $764_{(383-940)}$ | $576_{(156-876)}$ | $910_{(620-928)}$ | $557_{(270-887)}$ | $853_{(580-906)}$ |
| | | Reach Bottom Right | $0_{(0-0)}$ | $0_{(0-0)}$ | $0_{(0-18)}$ | $0_{(0-0)}$ | $0_{(0-0)}$ | $0_{(0-0)}$ | $0_{(0-0)}$ | $0_{(0-0)}$ |
| | | Reach Bottom Left | $0_{(0-0)}$ | $799_{(537-806)}$ | $139_{(0-288)}$ | $0_{(0-0)}$ | $0_{(0-0)}$ | $0_{(0-0)}$ | $0_{(0-0)}$ | $0_{(0-0)}$ |
| | Jaco | Reach Top Right | $17_{(10-29)}$ | $0_{(0-0)}$ | $0_{(0-0)}$ | $8_{(1-21)}$ | $2_{(0-8)}$ | $5_{(2-11)}$ | $0_{(0-0)}$ | $9_{(6-19)}$ |
| | | Reach Top Left | $10_{(5-18)}$ | $0_{(0-0)}$ | $0_{(0-0)}$ | $0_{(0-1)}$ | $2_{(0-5)}$ | $7_{(0-14)}$ | $27_{(2-53)}$ | $0_{(0-0)}$ |
| | | Reach Bottom Right | $17_{(11-33)}$ | $0_{(0-0)}$ | $0_{(0-0)}$ | $3_{(2-7)}$ | $4_{(2-13)}$ | $5_{(2-14)}$ | $0_{(0-0)}$ | $11_{(1-36)}$ |
| | | Reach Bottom Left | $2_{(0-12)}$ | $0_{(0-0)}$ | $0_{(0-0)}$ | $2_{(0-3)}$ | $10_{(5-19)}$ | $5_{(0-8)}$ | $15_{(0-39)}$ | $9_{(4-16)}$ |

Table 7: **Full dataset experimental results on ExORL.** For each dataset-domain pair, we report the score at the step for which the all-task IQM is maximised when averaging across 5 seeds, and the constituent task scores at that step. Bracketed numbers represent the 95% confidence interval obtained by a stratified bootstrap.

| Dataset | Domain | Task | FB | VC-FB (ours) | MC-FB (ours) |
|---|---|---|---|---|---|
| RND | Walker | Walk | $821_{(758-883)}$ | $864_{(850-879)}$ | $792_{(728-857)}$ |
| | | Stand | $928_{(925-930)}$ | $878_{(854-903)}$ | $873_{(812-934)}$ |
| | | Run | $281_{(242-320)}$ | $351_{(328-374)}$ | $343_{(323-366)}$ |
| | | Flip | $525_{(452-598)}$ | $542_{(513-571)}$ | $598_{(538-657)}$ |
| | Quadruped | Stand | $957_{(952-963)}$ | $863_{(777-950)}$ | $949_{(939-958)}$ |
| | | Roll Fast | $574_{(553-599)}$ | $512_{(471-553)}$ | $565_{(555-575)}$ |
| | | Roll | $920_{(895-944)}$ | $831_{(741-920)}$ | $890_{(874-906)}$ |
| | | Jump | $736_{(721-751)}$ | $630_{(570-690)}$ | $705_{(703-707)}$ |
| | | Escape | $94_{(63-125)}$ | $59_{(50-68)}$ | $66_{(56-86)}$ |
| | Maze | Reach Top Right | $0_{(0-0)}$ | $425_{(153-698)}$ | $270_{(9-533)}$ |
| | | Reach Top Left | $612_{(313-911)}$ | $454_{(138-769)}$ | $773_{(611-934)}$ |
| | | Reach Bottom Right | $0_{(0-0)}$ | $0_{(0-0)}$ | $0_{(0-0)}$ |
| | | Reach Bottom Left | $268_{(0-536)}$ | $270_{(2-539)}$ | $1_{(0-2)}$ |
| | Jaco | Reach Top Right | $48_{(39-56)}$ | $24_{(0-47)}$ | $51_{(23-79)}$ |
| | | Reach Top Left | $23_{(6-40)}$ | $14_{(4-25)}$ | $20_{(7-33)}$ |
| | | Reach Bottom Right | $60_{(56-65)}$ | $5_{(0-10)}$ | $47_{(15-79)}$ |
| | | Reach Bottom Left | $27_{(12-42)}$ | $88_{(33-143)}$ | $20_{(9-30)}$ |
| RANDOM | Walker | Walk | $148_{(70-225)}$ | $145_{(109-182)}$ | $129_{(80-178)}$ |
| | | Stand | $318_{(281-355)}$ | $255_{(202-308)}$ | $285_{(262-309)}$ |
| | | Run | $51_{(45-60)}$ | $47_{(44-50)}$ | $45_{(36-55)}$ |
| | | Flip | $57_{(49-67)}$ | $83_{(49-117)}$ | $103_{(65-140)}$ |
| | Quadruped | Stand | $417_{(393-453)}$ | $295_{(165-424)}$ | $210_{(153-267)}$ |
| | | Roll Fast | $110_{(51-170)}$ | $271_{(252-290)}$ | $215_{(139-292)}$ |
| | | Roll | $231_{(116-346)}$ | $154_{(53-255)}$ | $303_{(160-530)}$ |
| | | Jump | $287_{(123-450)}$ | $67_{(44-90)}$ | $164_{(135-194)}$ |
| | | Escape | $10_{(6-14)}$ | $7_{(4-10)}$ | $12_{(9-17)}$ |
| | Maze | Reach Top Right | $0_{(0-0)}$ | $0_{(0-0)}$ | $0_{(0-0)}$ |
| | | Reach Top Left | $309_{(4-615)}$ | $317_{(5-629)}$ | $307_{(0-614)}$ |
| | | Reach Bottom Right | $0_{(0-0)}$ | $0_{(0-0)}$ | $0_{(0-0)}$ |
| | | Reach Bottom Left | $0_{(0-0)}$ | $0_{(0-0)}$ | $0_{(0-0)}$ |
| | Jaco | Reach Top Right | $1_{(1-1)}$ | $2_{(0-4)}$ | $5_{(0-10)}$ |
| | | Reach Top Left | $50_{(0-100)}$ | $9_{(0-18)}$ | $16_{(0-31)}$ |
| | | Reach Bottom Right | $0_{(0-0)}$ | $15_{(5-25)}$ | $21_{(0-42)}$ |
| | | Reach Bottom Left | $3_{(1-6)}$ | $18_{(2-34)}$ | $1_{(0-3)}$ |
| DIAYN | Walker | Walk | $459_{(278-652)}$ | $536_{(305-766)}$ | $519_{(315-722)}$ |
| | | Stand | $478_{(463-494)}$ | $447_{(422-472)}$ | $517_{(433-602)}$ |
| | | Run | $87_{(81-93)}$ | $84_{(78-89)}$ | $87_{(65-110)}$ |
| | | Flip | $235_{(151-319)}$ | $251_{(151-352)}$ | $301_{(213-388)}$ |
| | Quadruped | Stand | $763_{(725-814)}$ | $432_{(176-688)}$ | $804_{(756-851)}$ |
| | | Roll Fast | $497_{(480-514)}$ | $293_{(179-407)}$ | $495_{(491-498)}$ |
| | | Roll | $767_{(726-808)}$ | $350_{(152-650)}$ | $761_{(736-786)}$ |
| | | Jump | $628_{(587-669)}$ | $234_{(89-379)}$ | $608_{(594-622)}$ |
| | | Escape | $65_{(62-69)}$ | $21_{(14-27)}$ | $67_{(55-79)}$ |
| | Maze | Reach Top Right | $0_{(0-0)}$ | $0_{(0-0)}$ | $0_{(0-0)}$ |
| | | Reach Top Left | $654_{(565-742)}$ | $928_{(907-950)}$ | $814_{(725-903)}$ |
| | | Reach Bottom Right | $0_{(0-0)}$ | $0_{(0-0)}$ | $8_{(0-16)}$ |
| | | Reach Bottom Left | $169_{(0-506)}$ | $7_{(0-14)}$ | $49_{(0-98)}$ |
| | Jaco | Reach Top Right | $4_{(2-7)}$ | $10_{(5-15)}$ | $4_{(1-8)}$ |
| | | Reach Top Left | $5_{(1-10)}$ | $1_{(0-2)}$ | $2_{(0-3)}$ |
| | | Reach Bottom Right | $9_{(4-13)}$ | $6_{(3-8)}$ | $7_{(0-14)}$ |
| | | Reach Bottom Left | $25_{(2-47)}$ | $12_{(6-18)}$ | $25_{(3-47)}$ |

Table 8: **Aggregate zero-shot performance on ExORL for all evaluation statistics recommended by [1].** VC-FB outperforms all methods across all evaluation statistics. ↑ means a higher score is better; ↓ means a lower score is better. Note that the optimality gap is large because we set $\gamma = 1000$ and for many dataset-domain-tasks the maximum achievable score is far from 1000.

| Statistic | SF-LAP | GC-IQL | FB | CQL | **MC-FB (ours)** | **VC-FB (ours)** |
|---|---|---|---|---|---|---|
| IQM ↑ | 92 | 95 | 99 | 128 | 136 | **148** |
| Mean ↑ | 87 | 126 | 118 | 138 | 142 | **154** |
| Median ↑ | 108 | 104 | 104 | 132 | 133 | **144** |
| Optimality Gap ↓ | 0.92 | 0.88 | 0.89 | 0.87 | 0.86 | **0.84** |

Table 9: **D4RL experimental results.** For each dataset-domain pair, we report the score at the step for which the IQM is maximised when averaging across 3 seeds. Bracketed numbers represent the 95% confidence interval obtained by a stratified bootstrap..

| Domain | Dataset | CQL | SF-LAP | FB | **VC-FB (ours)** | **MC-FB (ours)** |
|---|---|---|---|---|---|---|
| HalfCheetah | medium | 36 (36-36) | 4 (1-7) | 3 (-1-8) | 39 (38-40) | 32 (28-36) |
| | medium-expert | 43 (37-51) | 26 (20-32) | 3 (0-6) | 27 (24-28) | 24 (21-30) |
| | medium-replay | 42 (42-42) | 29 (29-30) | 7 (2-11) | 31 (26-36) | 20 (18-22) |
| Walker2d | medium | 70 (68-73) | 7 (1-13) | 7 (0-14) | 42 (37-45) | 36 (33-40) |
| | medium-expert | 102 (97-107) | 15 (0-31) | 3 (1-5) | 82 (72-92) | 74 (66-77) |
| | medium-replay | 13 (11-14) | 11 (8-15) | 12 (7-17) | 20 (19-21) | 22 (19-24) |
| All | All | 48 | 15 | 5 | 34 | 28 |

## D  Value Conservative Universal Successor Features

In this section, we develop *value conservative* regularisation for use by Universal Successor Features (USF) [5, 7], the primary alternative to FB for zero-shot RL.

Recall from Section 2 that successor features require a state-feature mapping $\varphi : \mathcal{S} \rightarrow \mathbb{R}^d$ which is usually obtained by some representation learning method [5]. *Universal* successor features are the expected discounted sum of these features, starting in state $s_0$, taking action $a_0$ and following the task-dependent policy $\pi_z$ thereafter

$$\psi(s_0, a_0, z) := \mathbb{E}\left[\sum_{t \geq 0} \gamma^t \varphi(s_{t+1}) | s_0, a_0, \pi_z\right]. \tag{21}$$

USFs satisfy a Bellman equation [7] and so can be trained using TD-learning on the Bellman residuals:

$$\mathcal{L}_{\text{SF}} = \mathbb{E}_{(s_t,a_t,s_{t+1})\sim\mathcal{D},z\sim\mathcal{Z}} \left(\psi(s_t, a_t, z)^\top z - \varphi(s_{t+1})^\top z - \gamma\bar{\psi}(s_{t+1}, \pi_z(s_{t+1}), z)^\top z\right)^2, \tag{22}$$

where $\bar{\psi}$ is a lagging target network updated via Polyak averaging, and $\mathcal{Z}$ is identical to that used for FB training (Appendix B.1.2). As with FB representations, the policy maximises the $Q$ function defined by $\psi$:

$$\pi_z(s) := \text{argmax}_a \psi(s, a, z)^\top z, \tag{23}$$

and for continuous state and action spaces is trained in an actor critic formulation. Like FB, USF training requires next action samples $a_{t+1} \sim \pi_z(s_{t+1})$ for the TD targets. We therefore expect SFs to suffer the same failure mode discussed in Section 3 (OOD state-action value overestimation) and to benefit from the same remedial measures (value conservatism). Training *value-conservative successor features* (VC-SF) amounts to substituting the USF $Q$ function definition and loss for FB's in Equation 11:

$$\mathcal{L}_{\text{VC-SF}} = \alpha \cdot \left( \mathbb{E}_{s \sim \mathcal{D}, a \sim \mu(a|s), z \sim \mathcal{Z}}[\psi(s,a,z)^\top z] - \mathbb{E}_{(s,a) \sim \mathcal{D}, z \sim \mathcal{Z}}[\psi(s,a,z)^\top z] \right) + \mathcal{L}_{\text{SF}}. \qquad (24)$$

Both the maximum value approximator $\mu(a|s)$ (Equation 17, Section B.1.3) and $\alpha$-tuning (Equation 18, Section B.1.4) can be extracted identically to the FB case with any occurrence of $F(s,a,z)^\top z$ substituted with $\psi(s,a,z)^\top z$. As USFs do not predict successor measures we cannot formulate measure-conservative USFs.

# E    Negative Results

In this section we provide detail on experiments we attempted, but which did not provide results significant enough to be included in the main body.

## E.1    Downstream Finetuning

If we relax the zero-shot requirement, could pre-trained conservative FB representations be finetuned on new tasks or domains? Base CQL models have been finetuned effectively on unseen tasks using both online and offline data [41], and we had hoped to replicate similar results with VC-FB and MC-FB. We ran offline and online finetuning experiments and provide details on their setups and results below. All experiments were conducted on the Walker domain.

**Offline finetuning.** We considered a setting where models are trained on a low quality dataset initially, before a high quality dataset becomes available downstream. We used models trained on the RANDOM-100k dataset and finetuned them on both the full RND and RND-100k datasets, with models trained from scratch used as our baseline. Finetuning involved the usual training protocol as described in Algorithm 1, but we limited the number of learning steps to 250k.

We found that though performance improved during finetuning, it improved no quicker than the models trained from scratch. This held for both the full RND and RND-100k datasets. We conclude that the parameter initialisation delivered after training on a low quality dataset does not obviously expedite learning when high quality data becomes available.

**Online finetuning.** We considered the online finetuning setup where a trained representation is deployed in the target environment, required to complete a specified task, and allowed to collect a replay buffer of reward-labelled online experience. We followed a standard online RL protocol where a batch of transitions was sampled from the online replay buffer after each environment step for use in updating the model's parameters. We experimented with fixing $z$ to the target task during in the actor updates (Line 16, Algorithm 1), but found it caused a quick, irrecoverable collapse in actor performance. This suggested uniform samples from $\mathcal{Z}$ provide a form of regularisation. We granted the agents 500k steps of interaction for online finetuning.

We found that performance never improved beyond the pre-trained (init) performance during finetuning. We speculated that this was similar to the well-documented failure mode of online finetuning of CQL [59], namely taking sub-optimal actions in the real env, observing unexpectedly high reward, and updating their policy toward these sub-optimal actions. But we note that FB representations do not update w.r.t observed rewards, and so conclude this cannot be the failure mode. Instead it seems likely that FB algorithms cannot

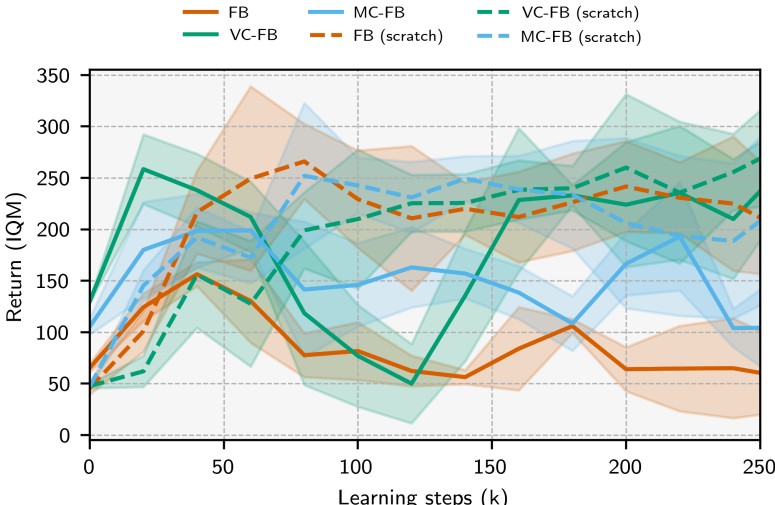

Figure 9: **Learning curves for methods finetuned on the full RND dataset.** Solid lines represent base models trained on RANDOM-100k, then finetuned; dashed lines represent models trained from scratch. The finetuned models perform no better than models trained from scratch after 250k learning steps, suggesting model re-training is currently a better strategy than offline finetuning.

use the narrow, unexploratory experience obtained from attempting to perform a specific task to improve model performance.

We believe resolving issues associated with finetuning conservative FB algorithms once the zero-shot requirement is relaxed is an important future direction and hope that details of our negative attempts to this end help facilitate future research.

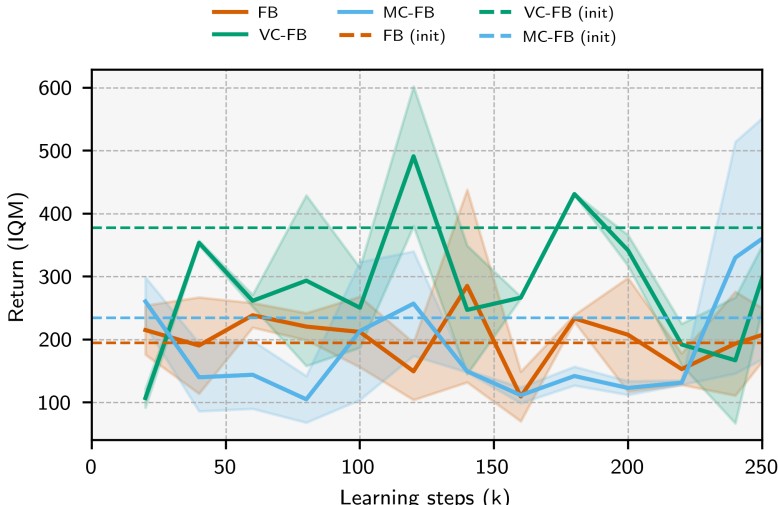

Figure 10: **Learning curves for online finetuning.** The performance at the end of pre-training (init performance) is plotted as a dashed line for each method. None of the methods consistently outperform their init performance after 250k online transitions.

## F   Learning Curves & Hyperparameter Sensitivity

**ExORL Learning Curves.** We report the learning curves for all zero-shot RL methods and CQL in Figures 11, 12, and 13. For all domains except Jaco, the y-axis limit is fixed at 1000 as that is the maximum score achievable in the DeepMind Control Suite. For Jaco-related figures, the y-axis limits is fixed at 100 as no method achieves a score higher than this.

**Hyperparameter Sensitivity.** We report the sensitivity of VC-FB and MC-FB to the choice of two new hyperparameters: conservative budget $\tau$ and action samples per policy $N$ on the ExORL benchmark. Figure 14 plots the sensitivity of VC-FB to the choice of $\tau$ on Walker and Maze domains across RND and RANDOM datasets. Figure 15 plots the sensitivity of MC-FB to the choice of $\tau$ on Walker and Maze domains across RND and RANDOM datasets. Figure 16 plots the sensitivity of MC-FB to the choice of $N$ on Walker and Maze domains across RND and RANDOM datasets.

We further explore the sensitivity of VC-FB performance on Walker2D from the D4RL benchmark w.r.t. the choice of conservative budget $\tau$. Figure 17 plots this relationship when trained on the "medium-expert" dataset from D4RL.

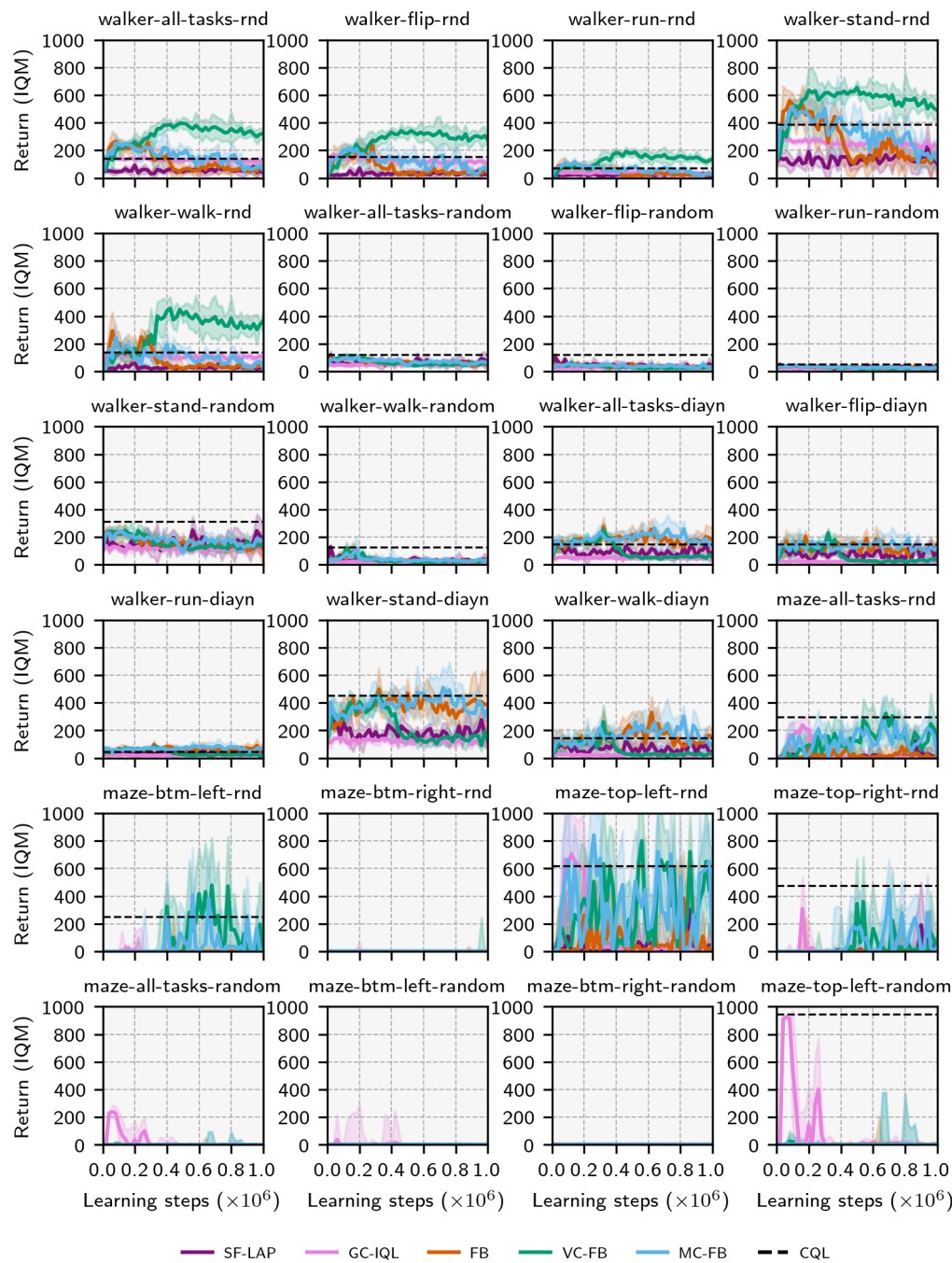

Figure 11: **Learning Curves (1/3)**. Models are evaluated every 20,000 timesteps where we perform 10 rollouts and record the IQM. Curves are the IQM of this value across 5 seeds; shaded areas are one standard deviation.

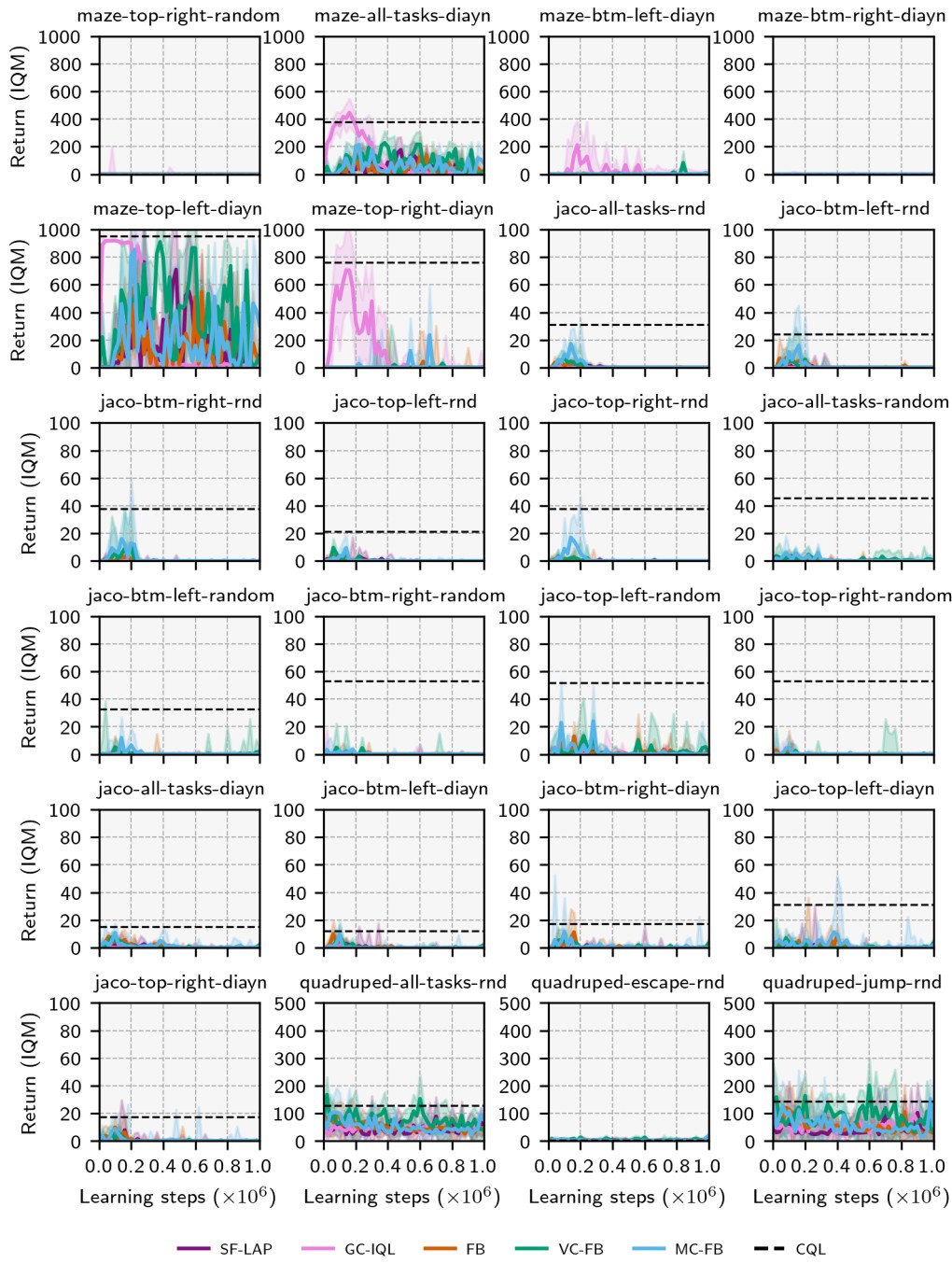

Figure 12: **Learning Curves (2/3)**. Models are evaluated every 20,000 timesteps where we perform 10 rollouts and record the IQM. Curves are the IQM of this value across 5 seeds; shaded areas are one standard deviation.

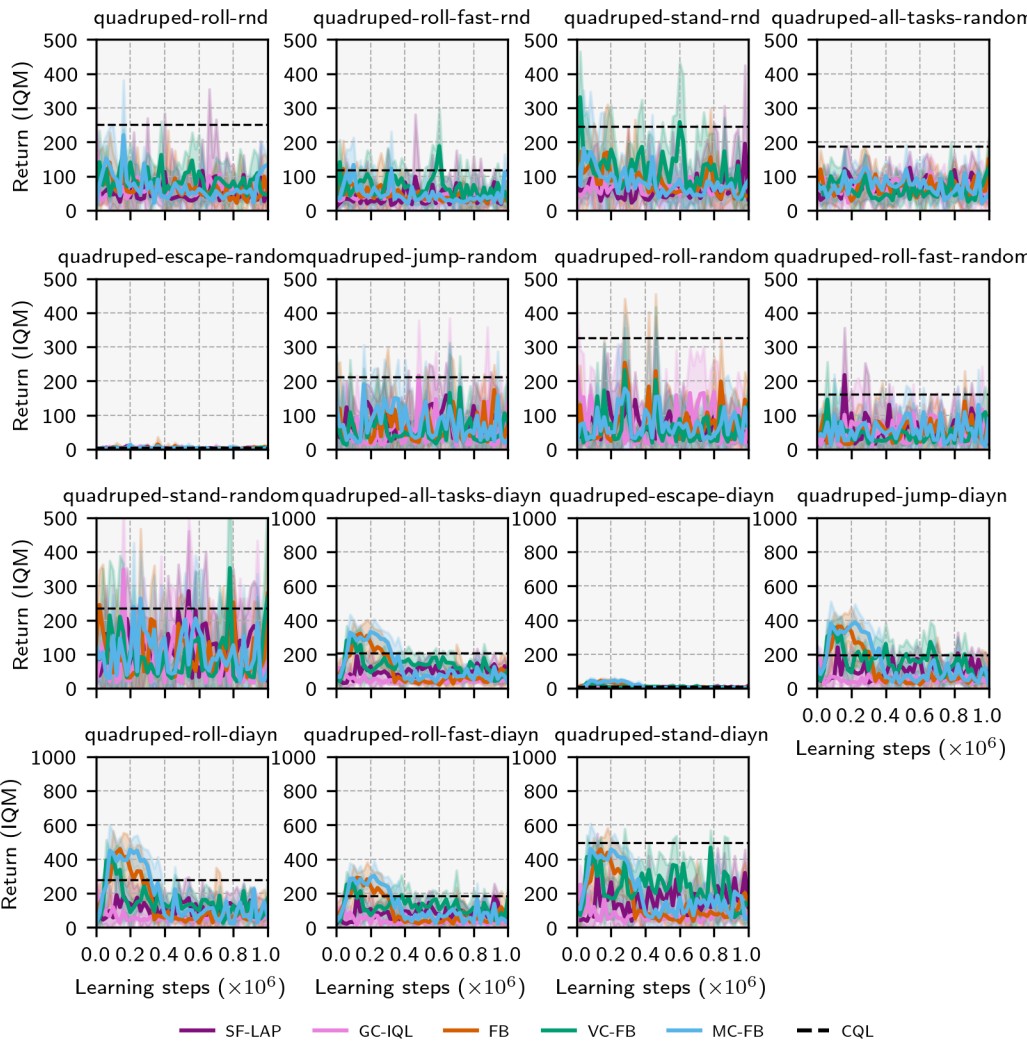

Figure 13: **Learning Curves (3/3)**. Models are evaluated every 20,000 timesteps where we perform 10 rollouts and record the IQM. Curves are the IQM of this value across 5 seeds; shaded areas are one standard deviation.

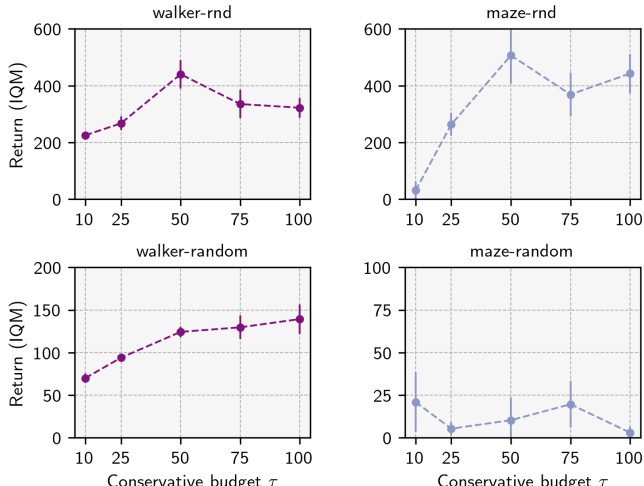

Figure 14: **VC-FB sensitivity to conservative budget $\tau$ on Walker and Maze**. Top: RND dataset; bottom: RANDOM dataset. Maximum IQM return across the training run averaged over 3 random seeds

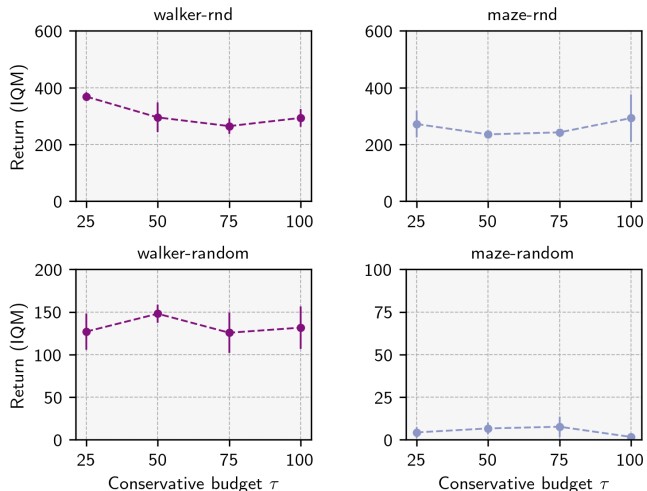

Figure 15: **MC-FB sensitivity to conservative budget $\tau$ on Walker and Maze**. Top: RND dataset; bottom: RANDOM dataset. Maximum IQM return across the training run averaged over 3 random seeds

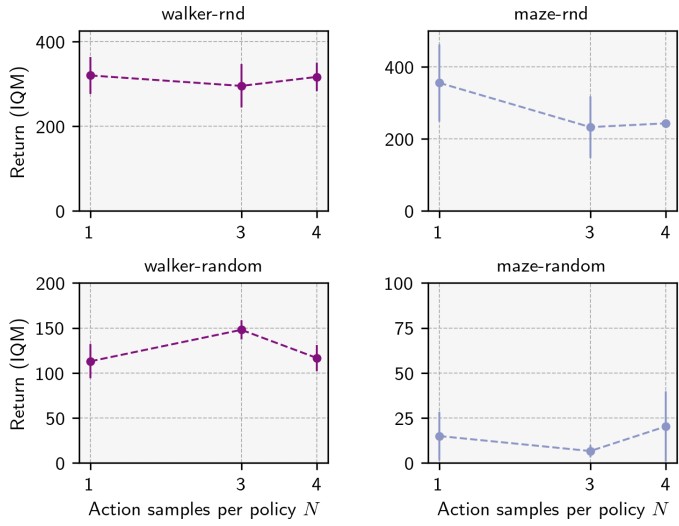

Figure 16: **MC-FB sensitivity to action samples per policy** $N$ **on** **Walker** **and** **Maze**. Top: RND dataset; bottom: RANDOM dataset. Maximum IQM return across the training run averaged over 3 random seeds.

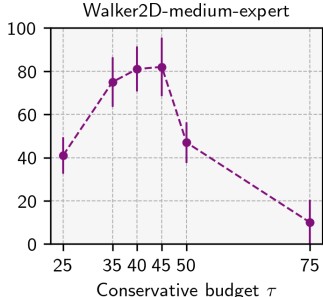

Figure 17: **VC-FB sensitivity to choice of conservative budget** $\tau$ **on** **Walker2D** from the D4RL benchmark.

# G   Code Snippets

## G.1   Update Step

```python
def update_fb(
    self,
    observations: torch.Tensor,
    actions: torch.Tensor,
    next_observations: torch.Tensor,
    discounts: torch.Tensor,
    zs: torch.Tensor,
    step: int,
) -> Dict[str, float]:
    """
    Calculates the loss for the forward-backward representation network.
    Loss contains two components:
        1. Forward-backward representation (core) loss: a Bellman update
           on the successor measure (equation 24, Appendix B)
        2. Conservative loss: penalises out-of-distribution actions
    Args:
        observations: observation tensor of shape [batch_size, observation_length]
        actions: action tensor of shape [batch_size, action_length]
        next_observations: next observation tensor of
                           shape [batch_size, observation_length]
        discounts: discount tensor of shape [batch_size, 1]
        zs: policy tensor of shape [batch_size, z_dimension]
        step: current training step
    Returns:
        metrics: dictionary of metrics for logging
    """

    # update step common to all FB models
    (
        core_loss,
        core_metrics,
        F1,
        F2,
        B_next,
        M1_next,
        M2_next,
        _,
        _,
        actor_std_dev,
    ) = self._update_fb_inner(
        observations=observations,
        actions=actions,
        next_observations=next_observations,
        discounts=discounts,
        zs=zs,
        step=step,
    )

    # calculate MC or VC penalty
    if self.mcfb:
        (
            conservative_penalty,
            conservative_metrics,
        ) = self._measure_conservative_penalty(
            observations=observations,
            next_observations=next_observations,
            zs=zs,
            actor_std_dev=actor_std_dev,
            F1=F1,
            F2=F2,
            B_next=B_next,
            M1_next=M1_next,
            M2_next=M2_next,
        )
    # VCFB
    else:
        (
            conservative_penalty,
            conservative_metrics,
        ) = self._value_conservative_penalty(
            observations=observations,
            next_observations=next_observations,
            zs=zs,
```

```
74                    actor_std_dev=actor_std_dev ,
75                    F1=F1 ,
76                    F2=F2 ,
77                )
78
79            # tune alpha from conservative penalty
80            alpha , alpha_metrics = self._tune_alpha (
81                conservative_penalty=conservative_penalty
82            )
83            conservative_loss = alpha * conservative_penalty
84
85            total_loss = core_loss + conservative_loss
86
87            # step optimiser
88            self.FB_optimiser.zero_grad ( set_to_none=True )
89            total_loss.backward ()
90            for param in self.FB.parameters ():
91                if param.grad is not None :
92                    param.grad.data.clamp_ (-1, 1)
93            self.FB_optimiser.step ()
94
95            return metrics
```

## G.2 Value-Conservative Penalty

```
1  def _value_conservative_penalty (
2          self ,
3          observations : torch.Tensor ,
4          next_observations : torch.Tensor ,
5          zs: torch.Tensor ,
6          actor_std_dev : torch.Tensor ,
7          F1: torch.Tensor ,
8          F2: torch.Tensor ,
9      ) -> torch.Tensor :
10         """
11         Calculates the value conservative penalty for FB.
12         Args:
13             observations: observation tensor of shape [batch_size , observation_length]
14             next_observations: next observation tensor of shape
15                                                  [batch_size , observation_length]
16             zs: task tensor of shape [batch_size , z_dimension]
17             actor_std_dev: standard deviation of the actor
18             F1: forward embedding no. 1
19             F2: forward embedding no. 2
20         Returns :
21             conservative_penalty: the value conservative penalty
22         """
23
24         with torch.no_grad ():
25             # repeat observations , next_observations , zs, and Bs
26             # we fold the action sample dimension into the batch dimension
27             # to allow the tensors to be passed through F and B; we then
28             # reshape the output back to maintain the action sample dimension
29             repeated_observations_ood = observations.repeat (
30                 self.ood_action_samples , 1, 1
31             ).reshape ( self.ood_action_samples * self.batch_size , -1)
32             repeated_zs_ood = zs.repeat ( self.ood_action_samples , 1, 1).reshape (
33                 self.ood_action_samples * self.batch_size , -1
34             )
35             ood_actions = torch.empty (
36                 size=( self.ood_action_samples * self.batch_size , self.action_length ),
37                 device=self._device ,
38             ).uniform_ (-1, 1)
39
40             repeated_observations_actor = observations.repeat (
41                 self.actor_action_samples , 1, 1
42             ).reshape ( self.actor_action_samples * self.batch_size , -1)
43             repeated_next_observations_actor = next_observations.repeat (
44                 self.actor_action_samples , 1, 1
45             ).reshape ( self.actor_action_samples * self.batch_size , -1)
46             repeated_zs_actor = zs.repeat ( self.actor_action_samples , 1, 1).reshape (
47                 self.actor_action_samples * self.batch_size , -1
48             )
49             actor_current_actions , _ = self.actor (
50                 repeated_observations_actor ,
51                 repeated_zs_actor ,
```

```
52              std=actor_std_dev,
53              sample=True,
54          )  # [actor_action_samples * batch_size, action_length]
55
56          actor_next_actions, _ = self.actor(
57              repeated_next_observations_actor,
58              z=repeated_zs_actor,
59              std=actor_std_dev,
60              sample=True,
61          )  # [actor_action_samples * batch_size, action_length]
62
63      # get Fs
64      ood_F1, ood_F2 = self.FB.forward_representation(
65          repeated_observations_ood, ood_actions, repeated_zs_ood
66      )  # [ood_action_samples * batch_size, latent_dim]
67
68      actor_current_F1, actor_current_F2 = self.FB.forward_representation(
69          repeated_observations_actor, actor_current_actions, repeated_zs_actor
70      )  # [actor_action_samples * batch_size, latent_dim]
71      actor_next_F1, actor_next_F2 = self.FB.forward_representation(
72          repeated_next_observations_actor, actor_next_actions, repeated_zs_actor
73      )  # [actor_action_samples * batch_size, latent_dim]
74      repeated_F1, repeated_F2 = F1.repeat(
75          self.actor_action_samples, 1, 1
76      ).reshape(self.actor_action_samples * self.batch_size, -1), F2.repeat(
77          self.actor_action_samples, 1, 1
78      ).reshape(
79          self.actor_action_samples * self.batch_size, -1
80      )
81      cat_F1 = torch.cat(
82          [
83              ood_F1,
84              actor_current_F1,
85              actor_next_F1,
86              repeated_F1,
87          ],
88          dim=0,
89      )
90      cat_F2 = torch.cat(
91          [
92              ood_F2,
93              actor_current_F2,
94              actor_next_F2,
95              repeated_F2,
96          ],
97          dim=0,
98      )
99
100     repeated_zs = zs.repeat(self.total_action_samples, 1, 1).reshape(
101         self.total_action_samples * self.batch_size, -1
102     )
103
104     # convert to Qs
105     cql_cat_Q1 = torch.einsum("sd, sd -> s", cat_F1, repeated_zs).reshape(
106         self.total_action_samples, self.batch_size, -1
107     )
108     cql_cat_Q2 = torch.einsum("sd, sd -> s", cat_F2, repeated_zs).reshape(
109         self.total_action_samples, self.batch_size, -1
110     )
111
112     cql_logsumexp = (
113         torch.logsumexp(cql_cat_Q1, dim=0).mean()
114         + torch.logsumexp(cql_cat_Q2, dim=0).mean()
115     )
116
117     # get existing Qs
118     Q1, Q2 = [torch.einsum("sd, sd -> s", F, zs) for F in [F1, F2]]
119
120     conservative_penalty = cql_logsumexp - (Q1 + Q2).mean()
121
122     return conservative_penalty
```

## G.3   Measure-Conservative Penalty

```
1 def _measure_conservative_penalty(
2         self,
```

```
3            observations: torch.Tensor,
4            next_observations: torch.Tensor,
5            zs: torch.Tensor,
6            actor_std_dev: torch.Tensor,
7            F1: torch.Tensor,
8            F2: torch.Tensor,
9            B_next: torch.Tensor,
10           M1_next: torch.Tensor,
11           M2_next: torch.Tensor,
12      ) -> torch.Tensor:
13           """
14           Calculates the measure conservative penalty.
15           Args:
16               observations: observation tensor of shape [batch_size, observation_length]
17               next_observations: next observation tensor of shape
18                                                [batch_size, observation_length]
19               zs: task tensor of shape [batch_size, z_dimension]
20               actor_std_dev: standard deviation of the actor
21               F1: forward embedding no. 1
22               F2: forward embedding no. 2
23               B_next: backward embedding
24               M1_next: successor measure no. 1
25               M2_next: successor measure no. 2
26           Returns:
27               conservative_penalty: the measure conservative penalty
28           """
29
30           with torch.no_grad():
31               # repeat observations, next_observations, zs, and Bs
32               # we fold the action sample dimension into the batch dimension
33               # to allow the tensors to be passed through F and B; we then
34               # reshape the output back to maintain the action sample dimension
35               repeated_observations_ood = observations.repeat(
36                   self.ood_action_samples, 1, 1
37               ).reshape(self.ood_action_samples * self.batch_size, -1)
38               repeated_zs_ood = zs.repeat(self.ood_action_samples, 1, 1).reshape(
39                   self.ood_action_samples * self.batch_size, -1
40               )
41               ood_actions = torch.empty(
42                   size=(self.ood_action_samples * self.batch_size, self.action_length),
43                   device=self._device,
44               ).uniform_(-1, 1)
45
46               repeated_observations_actor = observations.repeat(
47                   self.actor_action_samples, 1, 1
48               ).reshape(self.actor_action_samples * self.batch_size, -1)
49               repeated_next_observations_actor = next_observations.repeat(
50                   self.actor_action_samples, 1, 1
51               ).reshape(self.actor_action_samples * self.batch_size, -1)
52               repeated_zs_actor = zs.repeat(self.actor_action_samples, 1, 1).reshape(
53                   self.actor_action_samples * self.batch_size, -1
54               )
55               actor_current_actions, _ = self.actor(
56                   repeated_observations_actor,
57                   repeated_zs_actor,
58                   std=actor_std_dev,
59                   sample=True,
60               )  # [actor_action_samples * batch_size, action_length]
61
62               actor_next_actions, _ = self.actor(
63                   repeated_next_observations_actor,
64                   z=repeated_zs_actor,
65                   std=actor_std_dev,
66                   sample=True,
67               )  # [actor_action_samples * batch_size, action_length]
68
69           # get Fs
70           ood_F1, ood_F2 = self.FB.forward_representation(
71               repeated_observations_ood, ood_actions, repeated_zs_ood
72           )  # [ood_action_samples * batch_size, latent_dim]
73
74           actor_current_F1, actor_current_F2 = self.FB.forward_representation(
75               repeated_observations_actor, actor_current_actions, repeated_zs_actor
76           )  # [actor_action_samples * batch_size, latent_dim]
77           actor_next_F1, actor_next_F2 = self.FB.forward_representation(
78               repeated_next_observations_actor, actor_next_actions, repeated_zs_actor
79           )  # [actor_action_samples * batch_size, latent_dim]
80           repeated_F1, repeated_F2 = F1.repeat(
81               self.actor_action_samples, 1, 1
```

```
 82            ).reshape(self.actor_action_samples * self.batch_size, -1), F2.repeat(
 83                self.actor_action_samples, 1, 1
 84            ).reshape(
 85                self.actor_action_samples * self.batch_size, -1
 86            )
 87            cat_F1 = torch.cat(
 88                [
 89                    ood_F1,
 90                    actor_current_F1,
 91                    actor_next_F1,
 92                    repeated_F1,
 93                ],
 94                dim=0,
 95            )
 96            cat_F2 = torch.cat(
 97                [
 98                    ood_F2,
 99                    actor_current_F2,
100                    actor_next_F2,
101                    repeated_F2,
102                ],
103                dim=0,
104            )
105
106            cml_cat_M1 = torch.einsum("sd, td -> st", cat_F1, B_next).reshape(
107                self.total_action_samples, self.batch_size, -1
108            )
109            cml_cat_M2 = torch.einsum("sd, td -> st", cat_F2, B_next).reshape(
110                self.total_action_samples, self.batch_size, -1
111            )
112
113            cml_logsumexp = (
114                torch.logsumexp(cml_cat_M1, dim=0).mean()
115                + torch.logsumexp(cml_cat_M2, dim=0).mean()
116            )
117
118            conservative_penalty = cml_logsumexp - (M1_next + M2_next).mean()
119
120            return conservative_penalty
```

## G.4 $\alpha$ Tuning

```python
def _tune_alpha(
        self,
        conservative_penalty: torch.Tensor,
    ) -> torch.Tensor:
        """
        Tunes the conservative penalty weight (alpha) w.r.t. target penalty.
        Discussed in Appendix B.1.4
        Args:
            conservative_penalty: the current conservative penalty
        Returns:
            alpha: the updated alpha
        """

        # alpha auto-tuning
        alpha = torch.clamp(self.critic_log_alpha.exp(), min=0.0, max=1e6)
        alpha_loss = (
            -0.5 * alpha * (conservative_penalty - self.target_conservative_penalty)
        )

        self.critic_alpha_optimiser.zero_grad()
        alpha_loss.backward(retain_graph=True)
        self.critic_alpha_optimiser.step()
        alpha = torch.clamp(self.critic_log_alpha.exp(), min=0.0, max=1e6).detach()

        return alpha
```

