# OpenReview forum: "Zero-Shot Reinforcement Learning from Low Quality Data"
_NeurIPS.cc/2024/Conference — NeurIPS 2024 poster_

### Official Review · Reviewer_ZxeR · 2024-07-11

**Soundness:** 2
**Presentation:** 2
**Contribution:** 2
**Rating:** 3
**Confidence:** 2

**Summary:**

The work investigates methods for zero-shot reinforcement learning (RL) that can be trained on small, homogeneous datasets. This research is driven by the need to make zero-shot RL practical when large, heterogeneous datasets are unavailable. The authors identify the limitations of existing methods that overestimate out-of-distribution (OOD) state-action values when trained on low-quality datasets. They propose incorporating conservatism into zero-shot RL algorithms to mitigate these issues. Their experimental results demonstrate that conservative zero-shot RL methods outperform their non-conservative counterparts on low-quality datasets while maintaining competitive performance on high-quality datasets.

**Strengths:**

* Addressing Significant Issue: The paper attempts to address a significant gap in zero-shot RL by focusing on the challenges of using small, homogeneous datasets, which are more common in real-world applications.

* Improved Performance: The proposed conservative zero-shot RL methods consistently outperform non-conservative counterparts on
low-quality datasets and do not degrade performance on high-quality datasets.

**Weaknesses:**

1. **The problem definition**: I do not think the settings of this work is so-called Zero-shot RL since the work introduces another dataset. And the code link is invalid.
2. **Complexity of Implementation**: The introduction of conservatism increases the complexity of the algorithms, which may pose implementation challenges for practitioners.
3. **Limited Real-World Validation**: While the experiments are thorough, there is limited validation of the proposed methods in real-world settings, which could affect their applicability.
4. **Comparative Analysis**: The paper could benefit from a more in-depth comparative analysis with other state-of-the-art methods to highlight specific strengths and weaknesses.

**Questions:**

See in weakness.

**Limitations:**

See in weakness.

---

> ### Author Rebuttal · Authors · 2024-08-06
>
> Hi Reviewer ZxeR. See our responses to your comments below. We hope we can work with you to address any misunderstandings. Thank you in advance for your cooperation.
>
> **W1: The problem definition.** **I do not think the settings of this work is so-called Zero-shot RL since the work introduces another dataset.**
>
> We believe there is a critical misunderstanding in this question which we hope to clarify. We are very much performing zero-shot RL in this work. Zero-shot RL is about using a dataset of transitions collected from an environment to pre-train an agent such that it can return a performant policy for any downstream task in that environment. In our work, we are only changing the quality of this pre-training dataset. We are not introducing new datasets at test-time, and we are not fine-tuning agents on new data. We lay out our problem setting in lines 49-55, and this exactly matches the original zero-shot RL problem as proposed in the canonical work [1].
>
> **W2: Complexity of Implementation. The introduction of conservatism increases the complexity of the algorithms, which may pose implementation challenges for practitioners.**
>
> We disagree that our proposals would pose implementation challenges for practitioners. In lines 147-148 we stated: *“We emphasise these additions represent only a small increase in the number of lines required to implement existing methods.”* We provide code snippets in Appendix G where the reviewer can see that are proposals constitute a few 10s of extra lines of code on top of the vanilla FB implementation.
>
> **W3: Limited Real-World Validation.** **While the experiments are thorough, there is limited validation of the proposed methods in real-world settings, which could affect their applicability.**
>
> Whilst we agree that evaluating methods on simulated benchmarks raises questions about the real-world applicability of any method, we specifically chose D4RL as one of our testbeds because it is designed to mirror the difficulties/practicalities of real-world deployment. The D4RL paper describes the benchmark as being *“guided by key properties of datasets relevant to real-world applications of offline RL. (cf. abstract)”* And that *“the choice of tasks and datasets [are] motivated by properties reflected in real-world applications, such as narrow data distributions and undirected, logged behavior. (cf. section 7)”*.
>
> Because of these characteristics, D4RL has become the standard benchmark for evaluating (as closely as is feasible) the real-world applicability of offline RL algorithms, and our work is continuation of that tradition.
>
> **W4: Comparative Analysis.** **The paper could benefit from a more in-depth comparative analysis with other state-of-the-art methods to highlight specific strengths and weaknesses.**
>
> As discussed in section 4.2, we use both existing state of the art zero-shot RL methods (successor features with laplacian eigenfunctions and FB representations) as baselines. We also compare with the performance of TD3, the best performing single-task RL method on the ExORL benchmark, and CQL, the single-task equivalent of our method. To the best of our knowledge there are no other zero-shot RL methods for us to compare against.
>
> Our comparisons span 5 baseline algorithms, 11 datasets, 6 domains, 19 tasks, and 3/5 seeds for a total of 1086 individual runs. We believe Section 4 (Results) provides evidence of the strengths of our methods w.r.t to the existing state-of-the-art. We explore the weaknesses of our methods thoroughly with additional experiments in Section 5 (Limitations).
>
> **Code**
>
> Finally, we’re not sure what happened with our code link, it seems to have expired unexpectedly since we submitted this work. We’ve shared a new link with the AC (as per the guidelines) that they should share with you so you can review the code.
>
> **References**
>
> [1]  Ahmed Touati, Jérémy Rapin, and Yann Ollivier. Does zero-shot reinforcement learning exist? ICLR 2023

---

### Official Review · Reviewer_ZC95 · 2024-07-12

**Soundness:** 3
**Presentation:** 3
**Contribution:** 3
**Rating:** 6
**Confidence:** 3

**Summary:**

This work addresses zero-shot reinforcement learning (RL), focusing on training agents to perform tasks without explicit rewards during pre-training. The authors investigate the performance degradation that occurs with small, homogeneous datasets and propose conservative zero-shot RL algorithms inspired by single-task offline RL approaches. Experimental results demonstrate that the proposed methods improve performance on sub-optimal datasets without compromising effectiveness on large, diverse datasets.

**Strengths:**

* Zero-shot pretraining for RL is a crucial problem that can enhance generalization in downstream tasks.

* The proposed method is well-motivated, and the writing is clear.

* The experiments are thorough and effectively validate the proposed method.

**Weaknesses:**

* One weakness of the paper is the lack of a clear definition for "low-quality data." As shown in Figure 8, it might be more accurately described as "coverage" rather than quality. Providing a more explicit definition is recommended.

* Additionally, the comparison of baselines is incomplete, as it lacks an offline goal-conditioned baseline, which is highly relevant in the considered tasks. Including a representative offline GCRL baseline, such as GC-IQL, would strengthen the comparison.

* Some related works are missing, particularly in the offline GCRL domain. References [1][2][3][4] are pertinent as they also focus on learning from offline datasets without rewards and deploying for multiple goals. The didactic example in this paper is very similar to Figure 1 in a prior paper [3], which also convey similar information during zero-shot deployment for multiple goals. A discussion with the prior results would be beneficial. Furthermore, a very relevant offline zero-shot RL method [5] is neither cited nor discussed.


*References*

[1] Yang R, Lu Y, Li W, et al. Rethinking goal-conditioned supervised learning and its connection to offline rl. ICLR, 2022.

[2] Park S, Ghosh D, Eysenbach B, et al. Hiql: Offline goal-conditioned rl with latent states as actions. Advances in Neural Information Processing Systems, 2024.

[3] Yang R, Yong L, Ma X, et al. What is essential for unseen goal generalization of offline goal-conditioned rl?[C]//International Conference on Machine Learning. PMLR, 2023.

[4] Eysenbach B, Zhang T, Levine S, et al. Contrastive learning as goal-conditioned reinforcement learning[J]. Advances in Neural Information Processing Systems, 2022.

[5] Frans K, Park S, Abbeel P, et al. Unsupervised Zero-Shot Reinforcement Learning via Functional Reward Encodings[J]. arXiv preprint arXiv:2402.17135, 2024.

**Questions:**

* Could the authors provide a more explicit definition of "low-quality data"?

* It is recommended to compare the proposed method with a representative offline GCRL baseline.

* Including related works discussed above would make this study more comprehensive.

* Can this method also be applied to approaches based on successor features?

**Limitations:**

One limitation is that VC-FB and MC-FB do not outperform CQL on the D4RL benchmark; addressing this issue is left for future work.

---

> ### Author Rebuttal · Authors · 2024-08-06
>
> Hi Reviewer ZC95. Thanks very much for engaging with our paper and for the positive comments. See our response to your questions below.
>
> **Q1: Could the authors provide a more explicit definition of "low-quality data"?**
>
> To the best of our knowledge, there isn’t one metric that formalises the quality and/or coverage of an offline RL dataset. We think of datasets as having two independent characteristics: 1) diversity—how well the state-action space is covered, and 2) size—the absolute number of samples in the dataset. It’s possible to have high diversity and small size (RND-100k) or low diversity and large size (Random-Full). We used “quality” as an umbrella term for both size and diversity, though we agree we could have used “coverage”. If the reviewer feels strongly about this we could replace “low quality” with “low coverage” in the paper.
>
> **Q2-3: It is recommended to compare the proposed method with a representative offline GCRL baseline, and include related GCRL literature.**
>
> Zero-shot RL requires methods to generalise to both goal-reaching reward functions *and* “dense” reward functions (i.e. those that define locomotion tasks like Walker-Walk, Walker-Run etc.). We didn’t initially compare our methods to an offline GCRL baseline because such methods are, in principle, only capable of generalising to goal-reaching tasks. Indeed, it is not immediately clear how we would even define a goal for locomotion-type tasks. GCRL methods are therefore limited in a way zero-shot RL methods are not, so we didn’t think they represented a valid baseline.
>
> That said, we're keen to provide empirical evidence to support this explanation, and have run additional experiments to that end. We evaluated GC-IQL on our 4 ExORL domains when trained on the RND-100k buffer. In lieu of a well-defined goal state for the locomotion tasks, we used the state in $D_{\text{labelled}}$ with highest reward. We hypothesised that GC-IQL would perform similarly to our proposed methods on the goal-reaching tasks, and less well on the locomotion tasks. We also hypothesised that, because GC-IQL employs conservatism like VC-FB and MC-FB, it would handle the low-quality datasets better than vanilla FB and outperform it in aggregate, despite the naive goal definition for locomotion tasks. We report the aggregated performance of GC-IQL averaged across 5 seeds, compared with zero-shot RL methods based on FB in the table below.
>
> | Domain | Task | Task Type | FB | GC-IQL | VC-FB | MC-FB |
>  | --- | --- | --- | --- | --- | --- | --- |
>  | Walker | All Tasks | Locomotion | 266 (233–283)  | 218 (164 - 251) |  **396 (381–407)** | 252 (188–288) |
>   | Point-mass Maze | All Tasks | Goal-reaching | 102 (0–181) | **332 (190-436)** | 323 (177–412) | 270 (154–459) |
>  | Quadruped | All Tasks | Locomotion | 93 (69–137) | 85 (61-129) | **168 (104–201)** |  104 (38–212) |
>  | Jaco | All Tasks | Goal-reaching | 4 (1–6) | 15 (6-25) | 7 (3–12) | **17 (7–26)** |
>  | All Domains | All Tasks | - | 97 | 170 | **245** | 178 |
>
> We find that GC-IQL performs similarly to the VC-FB and MC-FB on the goal-reaching tasks as predicted. On the locomotion tasks it performs worse than all zero-shot methods (irrespective of whether they are conservative or not). This is presumably because the state with highest reward in $D_{\text{labelled}}$ is a poor proxy for the true, dense reward function.
>
> Thank you for this feedback, it has made us realise that we should clarify the difference between GCRL and zero-shot RL in the paper. We will add these results (and associated results for DIAYN and Random datasets), an explanation of the differences between GCRL and zero-shot RL, and the associated literature you cite in the revised manuscript.
>
> **Q4: Can this method also be applied to approaches based on successor features?**
>
> Yes, our value-conservative proposals are fully compatible with successor features. We mention this in line 99, and provide a derivation of value conservative successor features in Appendix D.

---

> > ### Comment · Reviewer_ZC95 · 2024-08-09
> >
> > Thank you for the responses. While most issues have been resolved, I think the first question remains unaddressed. The authors claim that "quality" is composed of "diversity" and "data size," which I disagree with. Does this imply that data with low diversity but large size is considered high quality? Additionally, "coverage" or "diversity" has been studied in previous works such as [1][2], whereas "quality" is often referred to as the average return of the trajectories. I suggest the authors reconsider their definition or terminology to ensure rigorousness of a good work.
> >
> > [1] Schweighofer K, Radler A, Dinu M C, et al. A Dataset Perspective on Offline Reinforcement Learning[J]. arXiv preprint arXiv:2111.04714, 2021.
> >
> > [2] Yarats D, Brandfonbrener D, Liu H, et al. Don't change the algorithm, change the data: Exploratory data for offline reinforcement learning[J]. arXiv preprint arXiv:2201.13425, 2022.

---

> ### Author Response · Authors · 2024-08-13
> **Response by authors**
>
> Thanks for pointing us toward this work. We currently cite [2], and use their proxy for dataset quality (downstream performance on certain tasks) to rank our datasets. However, we were unaware of [1], and the metrics they propose are helpful, although not directly applicable to our setting as we’ll explain, but we’re keen to try to use these ideas.
>
> In [1] they propose metrics for measuring the *exploitation (TQ)* and *exploration (SACo)* of the behaviour policy that creates the dataset. *TQ* is estimated as the mean reward of trajectories in the dataset w.r.t. a single downstream task of interest. Since we’re interested in generalising to any downstream task, in principle we’d need to calculate TQ for an infinite set of reward functions as specified by our task sampling distribution $\mathcal{Z}$ which is clearly intractable. As an alternative, we could calculate TQ w.r.t our test reward functions only, but our datasets are made up of state-action-next-state transition samples, not of full trajectories, so we can’t do this either. Consequently, TQ doesn’t quite fit our problem setting. However, we can measure *SACo;* the ratio of unique state-action pairs contained in a dataset w.r.t. some reference dataset. We used their codebase to do so w.r.t. an idealised dataset of 100k unique state-action transitions. See the metrics below, note we had to vary the number of discretisation bins to to get meaningful results across environments with different sizes of state and action spaces.
>
> |  | Walker (5 bins) | Point-mass Maze (25 bins) | Quadruped (3 bins) | Jaco (4 bins) |
> | --- | --- | --- | --- | --- |
> | RND-100k | 0.993 | 0.557 | 0.657 | 0.981 |
> | DIAYN-100k | 0.432 | 0.395 | 0.587 | 0.983 |
> | Random-100k | 0.477 | 0.036 | 0.526 | 0.801 |
> |  |  |  |  |  |
>
> Though this is a step in the right direction it is still not a complete picture. For that, we would need a TQ-like metric that is applicable to our setting. This has raised an interesting discussion which we’ll add to the paper to highlight the limitations of these existing approaches. We will add this table to Appendix B (Datasets), cite this work, update text in the main body to mention this method and to discuss its limitations.

---

### Official Review · Reviewer_5D6P · 2024-07-12

**Soundness:** 3
**Presentation:** 4
**Contribution:** 4
**Rating:** 7
**Confidence:** 2

**Summary:**

This paper identifies an overestimation problem of zero-shot RL algorithms, particularly in low data or low data quality settings. To address this, they propose to use a value conservative method to mitigate the overestimation (from CQL). They showcase that this effectively allows for reducing the overestimation, then empirically show performance improvements on low or bad quality data regimes, without impacting performance on usual regimes.

**Strengths:**

I am not an expert of zero short RL, but the paper seems sound and is mostly clearly written. The problem seems relevant as zero short RL from low quality data is relevant for robotic applications, as underlined by the authors. I like the first figure, that provides intuition, similar to the ones in DDQN's paper. The method is clearly motivated and explained, the experimental evaluation is clear, the figure and caption allow assessing the defined precise scientific questions. I do not see much flaw in this paper presentation and method, but my qualification is not too high.

**Weaknesses:**

**Provide a bit more background and intuition about zero-shot RL**. This is not really mandatory, but it would help a broader audience.

**Introduce the RANDOM and RND datasets**. Maybe an introduction of these datasets can help better grasp Figure 2 and 3, and would help the previous point.

**Questions:**

* What is the intuition behind the overestimation reduction of CQL ?

**Limitations:**

There is a dedicated limitation section that correctly addresses the limitations of the method.

---

> ### Author Rebuttal · Authors · 2024-08-06
>
> Hi Reviewer 5D6P. Thanks very much for engaging with our paper and for the positive comments. See our response to your questions below.
>
> **Q1: Provide a bit more background and intuition about zero-shot RL**.
>
> Thanks for the pointer. We’ll add the following text to help intuition after the formal introduction of zero-shot RL in lines 49-55.
>
> *“Intuitively, the zero-shot RL problem asks: is it possible to train an agent using a pre-collected dataset of transitions from an environment such that, at test time, it can return the optimal policy for *any* *task* in that environment without any further planning or learning?”*
>
> **Q2:** **Introduce the RANDOM and RND datasets (to help Figure 2)**.
>
> We introduce the datasets in lines 177-186 in Section 4.1 (after Figure 2), and provide more detail in Appendix A.2, including a visualisation of the state coverage on Point-mass Maze in Figure 8. We point to Appendix A.2 at the end of Figure 2’s caption in light of not being able to introduce the datasets earlier. We feel that, together, these provide good context on the datasets used in our study.
>
> **Q3: What is the intuition behind the overestimation reduction of CQL?**
>
> We provide (brief) intuition of CQL’s regularising effect in Lines 120-121, VC-FB’s regularising effect in Lines 124-125, and MC-FB’s regularising effect in Lines 131-134. We’ll summarise these again below for convenience.
>
> In offline RL $Q$ functions tend to over-value state-action pairs that aren’t in the dataset. Intuitively, CQL amends the $Q$ function’s loss such that the value of these state-action pairs are downweighted, whilst upweighting the value of state-action pairs that are in the dataset (Equation 10).  VC-FB does the same thing, but for all possible downstream tasks (Equation 11), rather than just one task as with CQL. MC-FB is slightly different in that the regulariser operates on expected future visitation counts (measures $M$), rather than expected values ($Q$ functions). The MC-FB loss function downweights the *probability* of reaching some future state $s_+$ from state-action pairs not in the dataset, whilst upweighting the probability of reaching some future state $s_+$ from a state-action pairs that are in the dataset.

---

> > ### Comment · Reviewer_5D6P · 2024-08-11
> > **Thank you very much for your clarifications**
> >
> > Thank you very much for your clarifications. I hope that they will help improve your manuscript, and make it more accessible to a broader audience.

---

### Official Review · Reviewer_qeSX · 2024-07-12

**Soundness:** 3
**Presentation:** 3
**Contribution:** 3
**Rating:** 7
**Confidence:** 3

**Summary:**

This work proposes modifications to Forward-Backward representations method. The modifications are aimed to improve the method's robustness to dataset quality. The vanilla FB suffers when the dataset does not cover the state space well, and overestimates the quality of actions. The authors show this with a simple example on point-mass. Two modifications are proposed, VC-FB and MC-FB, which take inspiration from CQL and make sure the FB representations are not too optimistic on the transitions not seen in the dataset.
VC-FB and MC-FB are then thoroughly tested on ExORL and D4RL. The authors show that VC-FB offers the best improvement on ExORL, although none of the proposed modifications beat CQL on D4RL benchmark. Nevertheless, in all settings the modifications greatly improve over vanilla FB.

**Strengths:**

- The paper is clearly written.
- The proposed modifications have clear mathematical motivation, similar to CQL
- Both the modifications improve over the vanilla FB representation, and even outperform CQL on ExORL
- The experiments are thorough and clear

**Weaknesses:**

- The novelty is limited: the authors take CQL's method for correcting overly optimal predictions and apply them to FB.

**Questions:**

-  How do the proposed modifications affect the performance of the model when the dataset is of good quality? Does it hurt the performance?

**Limitations:**

The authors have adequately described limitations and potential negative societal impact of their work.

---

> ### Author Rebuttal · Authors · 2024-08-06
>
> Hi Reviewer qeSX. Thanks very much for engaging with our paper and for the positive comments. See our response to your question below.
>
> **Q1: How do the proposed modifications affect the performance of the model when the dataset is of good quality? Does it hurt the performance?**
>
> We asked ourselves this question in Q3, Section 4, and responded to it on Lines 221-238, and with Figure 6, Table 1 and Table 8 (Appendix C). We’ll summarise our findings here for convenience.
>
> We compared the performance of our methods with vanilla FB on the full RND, DIAYN and Random datasets and found our methods showed superior performance on all of them (Table 1). We then ran a second experiment where we trained our methods and FB on different sizes (100k, 500k, 1m, 10m) of RND (the highest quality dataset). Again, we found our method outperformed vanilla FB on each of these, with the discrepancy in performance decreasing as we approached the fully sized dataset (Figure 6). As a result, we’re confident that conservative zero-shot RL methods do not trade better performance on low quality datasets for worse performance on the full, high-quality datasets.

---

> > ### Comment · Reviewer_qeSX · 2024-08-09
> >
> > Thank you for your response. My question regarding "good quality" dataset was rather about what would happen if the dataset you're pre-training on actually contains trajectories that solve the downstream task. In section 4.1 you say that you use offline TD3 performance on a given dataset as a proxy for dataset quality. My question is basically what will happen if you push the quality (as measured by TD3) even higher than you have so far. For example, you could take online TD3 replay buffer. Since there's not much time for you to collect the data and train on it, I'm just asking for you intuition about this.

---

> > > ### Author Response · Authors · 2024-08-11
> > > **Response from authors**
> > >
> > > Thanks for clarifying your question. We hint at the setting you describe when we train on the medium-expert dataset in our D4RL experiments (a dataset containing a mix of ~optimal and suboptimal trajectories for the downstream task). We report those results in Table 10 Appendix C. We found VC-FB and MC-FB were performant in this setting, unlike FB which fails catastrophically, but didn’t match the performance of CQL. Our intuition is that the success of our methods in this setting depends closely on the choice of $\tau$ which reflects how “conservative” our methods are w.r.t. OOD state-actions. If $\tau$ is too high, the methods are not conservative w.r.t. OOD state-actions and we get catastrophic failures akin to FB. If $\tau$ is too low the methods are too conservative about OOD state-actions and the policy struggles if it deviates from the ~optimal trajectories described by the dataset. Similar findings are discussed in the original CQL paper in Appendix F. Of course, this is far from perfect, and we’d like to look at more robust methods in future work.

---

### Decision · Program_Chairs · 2024-09-25

**Decision:**

Accept (poster)

**Comment:**

This paper addresses the challenges of zero-shot RL with low-quality data by proposing conservative algorithms that regularize value functions or dynamics predictions for out-of-distribution state-action pairs.

The paper was reviewed by four reviewers with three inclined to suggest for acceptance and one not. The majority of reviewers agree on the strengths, noting that (1) the proposed value conservative method is well motivated, (2) the paper is clearly written, and the approach is sound, and (3) the experiments are thorough and clear. Several weaknesses and questions were discussed during the rebuttal period, and I have not identified any major unresolved issues.

Considering these strengths, this paper is recommended for acceptance.
The authors are encouraged to utilize their constructive and detailed discussion with reviewers towards further raising the quality and impact of their work.